# SYMMETRIC PRUNING FOR LARGE LANGUAGE MODELS

## ABSTRACT

Popular post-training pruning methods such as Wanda (Sun et al., 2023) and RIA (Zhang et al., 2024b) are known for their simple, yet effective, designs that have shown exceptional empirical performance. Wanda optimizes performance through calibrated activations during pruning, while RIA emphasizes the relative, rather than absolute, importance of weight elements. Despite their practical success, a thorough theoretical foundation explaining these outcomes has been lacking. This paper introduces new theoretical insights that redefine the standard minimization objective for pruning, offering a deeper understanding of the factors contributing to their success. Our study extends beyond these insights by proposing complementary strategies that consider both input activations and weight significance. We validate these approaches through rigorous experiments, demonstrating substantial enhancements over existing methods. Furthermore, we introduce a novel training-free fine-tuning approach $R^2$-DSnoT that incorporates relative weight importance and a regularized decision boundary within a dynamic pruning-and-growing framework, significantly outperforming strong baselines and establishing a new state-of-the-art.

## 1 INTRODUCTION

Large Language Models (LLMs) (Zhang et al., 2022a; Touvron et al., 2023a;b; Javaheripi et al., 2023) have demonstrated remarkable capabilities across a variety of tasks. However, their extensive size often hinders practical deployment. Interest in LLM compression has surged in recent years, driven by the need to reduce model sizes while maintaining performance (Xiao et al., 2023; Frantar & Alistarh, 2023; Sun et al., 2023; Zhang et al., 2024b; Malinovskii et al., 2024). This paper focuses on LLM **post-training pruning (PTP)**, a prevalent method for reducing the footprint of pre-trained weights.

A common approach to pruning is magnitude-based pruning, where elements of each layer's weights with smaller absolute values are set to zero. In contrast, Wanda (Sun et al., 2023) introduced an innovative method that scales the weights by the activations of each layer, demonstrating promising performance on standard benchmarks. Building upon this, RIA (Zhang et al., 2024b) further improved the approach by evaluating the relative importance of each weight across its corresponding row and column before pruning. While their empirical results are encouraging, the underlying mechanisms remain poorly understood. This leads us to our first question:

*Can we provide theoretical support for post-training pruning methods and derive more efficient algorithms with minimal adaptations to the existing framework?*

To deepen our understanding of these popular PTP methods, we introduce a novel formulation—referred to as **Sym**metric **W**eight **A**nd **A**ctivation (SymWanda)—that aims to efficiently leverage *both* the input activation of a layer and the output for that layer. This symmetric and generalized approach provides theoretical insights into the mechanisms of established empirical methods such as Wanda and RIA.

Intrinsic PTP methods have demonstrated remarkable performance, as reflected by perplexity scores and zero-shot accuracy. However, their performance can degrade significantly when the sparsity ratio is high. This is due to the intrinsic reconstruction error between the pruned weights and the original pre-trained weights. Minimizing this reconstruction error is particularly important for effi-

cient post-training pruning. Beyond LLM pruning, we explore further fine-tuning to enhance model efficiency and performance. This brings us to our second problem:

*Can we fine-tune pruned LLMs without further training and outperforms state-of-the-art methods with minimal effort?*

**Dynamic sparse training (DST)** has gained attention for selectively updating and maintaining a subset of network parameters throughout the training process while dynamically adapting the sparse topology through weight operations. Its proven efficiency in enabling effective training suggests DST could be a promising approach for fine-tuning LLMs in an efficient manner. However, DST inherently requires backpropagation to train subnetworks, and its effectiveness heavily depends on a sufficient number of weight updates (Liu et al., 2021).

Interestingly, the pruning-and-growing step within DST offers a training-free methodology, where sparse mask adaptation is based solely on weight properties such as magnitude (Mocanu et al., 2018). This opens up a potential alternative for addressing the challenge: Instead of relying on computationally intensive backpropagation for fine-tuning sparse LLMs, we can explore the iterative updating of sparse masks in a training-free manner. Motivated by this insight, we focus on training-free fine-tuning approaches.

DSnoT (Zhang et al., 2023) introduced a straightforward yet effective method for pruning and growing weights using their values and statistical metrics (e.g., expectation and variance) for each ongoing pruning row. Inspired by Wanda, DSnoT achieves simplicity but falls short of fully leveraging relative weight information, particularly in scenarios where weight distributions are highly non-uniform and contain many outliers (Zhang et al., 2024b). To address these limitations, we propose incorporating relative weight importance into the growing criterion design. Furthermore, we observe that directly optimizing for reconstruction error is suboptimal. To improve performance, we introduce a regularization term that relaxes the decision boundary. Our new designs demonstrate significant efficiency and consistently achieve promising performance, paving the way for more effective and computationally feasible fine-tuning methods for sparse LLMs.

Our **contributions** are summarized as follows:

- We propose a novel formulation, SymWanda, which minimizes the impact of pruning on both input activations and output influences of weights. This approach provides theoretical insights into the empirical successes of methods such as Wanda and RIA.

- Building on this formulation, we introduce a series of innovative pruning strategies. Extensive experiments validate the effectiveness of our methods. Notably, we incorporate an efficient stochastic approach for manipulating relative importance, which achieves superior performance with highly reduced sampling cost.

- We present a novel training-free fine-tuning method $R^2$-DSnoT that leverages relative weight importance and a regularized decision boundary within a pruning-and-growing framework. This approach significantly outperforms strong baselines, achieving remarkable results.

## 2  RELATED WORK

**Traditional model pruning.**  Pruning has emerged as a powerful strategy to compress and accelerate deep neural networks by removing redundant connections while preserving overall performance (Han et al., 2015; Frankle & Carbin, 2018; Hoefler et al., 2021). Early works introduced iterative pruning-and-retraining approaches, which iteratively identify unimportant weights, discard them, and retrain the resulting sparse network to recover accuracy (LeCun et al., 1989; Han et al., 2015). More recent dynamic sparse training techniques (Mocanu et al., 2018; Bellec et al., 2018; Lee et al., 2018; Mostafa & Wang, 2019) start from a sparse initialization and continuously prune and grow connections throughout training. These methods integrate sparsification into the training loop, yielding promising trade-offs between model size and performance. A prominent line of work has leveraged learnable thresholds to realize non-uniform sparsity (Kusupati et al., 2020) or combined magnitude-based pruning with periodic connectivity updates to regrow valuable weights (Evci et al., 2020; Lasby et al., 2023). However, most of these methods still rely on standard back-propagation over the full parameter set, which can be prohibitively expensive when scaling up to LLMs.

Table 1: Comparison of LLM post-training pruning algorithms.

| Algorithm | W? | Act.? | X | Y | $\mathbf{S}_{jk}$[a] | Comment |
|---|---|---|---|---|---|---|
| General Sym. | ✓ | ✓ | X | Y | $\|\mathbf{W}_{jk}\|(\|\mathbf{X}_{:j}\|_2 + \|\mathbf{Y}_{k:}\|_2)$ | Lemma 3.1 |
| Marginal | ✓ | ✗ | I | 0 | $\|\mathbf{W}_{jk}\|$ | - |
| Wanda | ✓ | ✓ | X | 0 | $\|\mathbf{W}_{jk}\|\,\|\mathbf{X}_{:j}\|_2$ | Corollary 3.2 |
| OWanda | ✓ | ✓ | 0 | Y | $\|\mathbf{W}_{jk}\|\,\|\mathbf{Y}_{k:}\|_2$ | Corollary 3.3 |
| Symmetric | ✓ | ✓ | $\mathbf{W}^T$ | $\mathbf{W}^T$ | $\|\mathbf{W}_{jk}\|\sqrt{\|\mathbf{W}_{j:}\|_2^2 + \|\mathbf{W}_{:k}\|_2^2}$ | Corollary 3.4 |
| RI (v1) | ✓ | ✗ | $t_j(1;\cdots;,1),\ t_j=(\sqrt{b}\,\|\mathbf{W}_{j:}\|_1)^{-1}$[a] | $s_k(1,\cdots,1),\ s_k=(\sqrt{c}\,\|\mathbf{W}_{:k}\|_1)^{-1}$ | $\|\mathbf{W}_{j:}\|_1^{-1}+\|\mathbf{W}_{:k}\|_1^{-1}$ | Theorem 3.5 |
| RI (v2) | ✓ | ✗ | $\mathrm{Diag}(\|\mathbf{W}_{1:}\|_1^{-1},\ldots,\|\mathbf{W}_{b:}\|_1^{-1})$ | $\mathrm{Diag}(\|\mathbf{W}_{:1}\|_1^{-1},\ldots,\|\mathbf{W}_{:c}\|_1^{-1})$ | $\|\mathbf{W}_{j:}\|_1^{-1}+\|\mathbf{W}_{:k}\|_1^{-1}$ | Theorem 3.5 |
| RIA | ✓ | ✓ | $\delta_{u=j}\delta_{v=p}\|\mathbf{C}_{:j}\|_2^\alpha\|\mathbf{W}_{j:}\|_1^{-1}$[c] | $\delta_{u=s}\delta_{v=k}\|\mathbf{C}_{:j}\|_2^\alpha\|\mathbf{W}_{:k}\|_1^{-1}$ | $\left(\|\mathbf{W}_{j:}\|_1^{-1}+\|\mathbf{W}_{:k}\|_1^{-1}\right)\|\mathbf{X}_{:j}\|_2^\alpha$ | Lemma 3.6 |
| General (diag.) | ✓ | ✓ | $\mathbf{A}\mathbf{D}_\mathbf{X}$[d] | $\mathbf{D}_\mathbf{Y}\mathbf{B}$ | $\|\mathbf{A}_{:j}\|_2\|\mathbf{W}_{j:}\|_1^{-1}+\|\mathbf{B}_{k:}\|_2\|\mathbf{W}_{:k}\|_1^{-1}$ | Lemma 3.7 |
| $\ell_p$-norm (v1) | ✓ | ✗[e] | $\|\mathbf{W}_{j:}\|_p^{-1}\cdot\|\mathbf{W}_{j:}\|_2^{-1}\cdot\mathbf{W}_{j:}^\top$ | $\|\mathbf{W}_{:k}\|_p^{-1}\cdot\|\mathbf{W}_{:k}\|_2^{-1}\cdot\mathbf{W}_{:k}^\top$ | $\|\mathbf{W}_{jk}\|(\|\mathbf{W}_{j:}\|_p^{-1}+\|\mathbf{W}_{:k}\|_p^{-1})$ | Lemma 3.8 |
| $\ell_p$-norm (v2) | ✓ | ✗ | $\|\mathbf{W}_{j:}\|_p^{-1}\cdot\mathbf{u}$ | $\|\mathbf{W}_{:k}\|_p^{-1}\cdot\mathbf{v}$ | $\|\mathbf{W}_{jk}\|(\|\mathbf{W}_{j:}\|_p^{-1}+\|\mathbf{W}_{:k}\|_p^{-1})$ | Lemma 3.9 |
| StochRIA | ✓ | ✗ | $\mathbf{1}_{\{i\in S_j\}}\left(\|\mathbf{W}_{j:S_j}\|_1\sqrt{\tau}\right)^{-1}$ | $\mathbf{1}_{\{i\in S_k\}}\left(\|\mathbf{W}_{S_k:k}\|_1\sqrt{\tau}\right)^{-1}$ | $\|\mathbf{W}_{jk}\|(\|\mathbf{W}_{j:S_j}\|_1^{-1}+\|\mathbf{W}_{S_k:k}\|_1^{-1})$ | Lemma 3.10 |

[a] Without loss of generality, we consider the elimination of a single weight, $\mathbf{W}_{jk}$. The detailed explanation can be found in Lemma 3.1 and Section 3.2.

[b] For simplicity, instead of displaying the entire matrices $\mathbf{X}$ and $\mathbf{Y}$, we present the columns $\mathbf{X}_{:j}$ and the rows $\mathbf{Y}_{k:}$. This design is employed in the algorithms RI, RIA, $\ell_p$-norm, and StochRIA.

[c] The Kronecker delta, denoted by $\delta_{ij}$, is a function of two indices $i$ and $j$ that equals 1 if $i = j$ and 0 otherwise.

[d] $\mathbf{D}_\mathbf{X}$ and $\mathbf{D}_\mathbf{Y}$ are the diagonal matrices associated with $\mathbf{W}$, as defined in Section 3.4.

[e] By default, for $\ell_p$-norm and StochRIA, we do not consider the input activation. However, the design is similar to the transition from RI to RIA, as described in Section 3.3.

**LLM post-training pruning.** The substantial computational demands of LLMs have raised the development of pruning methods tailored to reduce parameters counts without compromising performance (Li et al., 2023; Zhu et al., 2024). Among these methods, post-training pruning eliminates redundant parameters in a pre-training network without requiring resource-intensive fine-tuning. For instance, SparseGPT (Frantar & Alistarh, 2023) leverages second-order information to solve layer-wise reconstruction problems, supporting both unstructured and N:M structured sparsity (Zhou et al., 2021). Wanda (Sun et al., 2023) introduces a pruning metric that incorporates both weight magnitudes and corresponding input activations, achieving perplexity performance comparable to SparseGPT while surpassing simple magnitude-based pruning. The RIA method (Zhang et al., 2024b) builds on Wanda by considering relative weight importance, offering performance improvements at minimal additional cost. Moreover, DSnoT (Zhang et al., 2023) proposes pruning and regrowing weights based on statistical properties (e.g., mean and variance) in each pruning row, obviating the need for retraining.

## 3 SYMMETRIC WANDA

### 3.1 PREREQUISITES

Post-training pruning is defined as follows: consider a target sparsity ratio $\varepsilon \in [0,1)$, a set of calibration inputs $\mathbf{X} \in \mathbb{R}^{a\times b}$, and pre-trained weights $\mathbf{W} \in \mathbb{R}^{b\times c}$. For clarity in the mathematical framework, we abstract the dimensions of inputs and weights. Specifically, in the context of large language models, let $a := C_{\text{in}}$, $b := N \times L$, and $c \equiv C_{\text{out}}$, where $N$ and $L$ denote the batch size and sequence length, respectively. The objective is to identify an optimal pruned weight matrix $\widetilde{\mathbf{W}} \in \mathbb{R}^{b\times c}$ that minimizes:

$$f(\widetilde{\mathbf{W}}) := \|\mathbf{X}(\widetilde{\mathbf{W}} - \mathbf{W})\|_F^2, \qquad \text{(InpRecon)}$$

where the optimization challenge is:

$$\text{minimize } f(\widetilde{\mathbf{W}}) \ \ s.t. \ \ \text{Mem}(\widetilde{\mathbf{W}}) \leq (1 - \varepsilon)\text{Mem}(\mathbf{W}),$$

where $\text{Mem}(\cdot)$ denotes the memory consumption associated with a weight matrix, and (InpRecon) quantifies the input reconstruction error.

This formulation applies to various post-training compression techniques, including both pruning (Frantar & Alistarh, 2023; Sun et al., 2023; Zhang et al., 2024b) and quantization (Frantar et al., 2023; Egiazarian et al., 2024). Our focus here is specifically on post-training pruning.

## 3.2 Symmetric Wanda: New Formulations

Building upon the methods introduced in Wanda (Sun et al., 2023), which considered both weights and activations, and later improvements by RIA (Zhang et al., 2024b), which analyzed the relative importance of weights by summing over corresponding rows and columns, we provide new insights by redefining our optimization objective. Apart from the previous defined input calibration $\mathbf{X}$, we particularly introduce the output calibration $\mathbf{Y} \in \mathbb{R}^{c \times d}$. Considering both the input and output dependencies, we express the objective as:

$$g(\widetilde{\mathbf{W}}) := \|\mathbf{X}(\widetilde{\mathbf{W}} - \mathbf{W})\|_F + \|(\widetilde{\mathbf{W}} - \mathbf{W})\mathbf{Y}\|_F, \tag{Sym}$$

and propose to solve:

$$\text{minimize } g(\widetilde{\mathbf{W}}), \ \ s.t. \ \text{Mem}(\widetilde{\mathbf{W}}) \le (1 - \varepsilon)\text{Mem}(\mathbf{W}).$$

We refer to the method that utilizes the general matrix in (Sym) without instantiation as SymWanda, which is designed to minimize the reconstruction error affected by both the input $\mathbf{X}$ and the output $\mathbf{Y}$. It is important to note that this formulation employs *non-squared* Frobenius norms to facilitate better theoretical interpretations. A squared norm version is also provided in Appendix B for comparison. We elucidate the efficacy of both approaches and provide new theoretical insights into the performance advantages previously observed with Wanda and RIA.

**Lemma 3.1.** *Assume we aim to eliminate a single weight $\mathbf{W}_{jk}$, setting $\widetilde{\mathbf{W}}_{jk} = 0$ and keeping all other weights unchanged. The simplified expression for $g(\widetilde{\mathbf{W}})$ becomes:*

$$g(\widetilde{\mathbf{W}}) = |\mathbf{W}_{jk}| \left( \|\mathbf{X}_{:j}\|_2 + \|\mathbf{Y}_{k:}\|_2 \right) := \mathbf{S}_{jk}, \tag{1}$$

*where $\mathbf{X}_{:j}$ and $\mathbf{Y}_{k:}$ represent the $j$-th column and $k$-th row of $\mathbf{X}$ and $\mathbf{Y}$, respectively.*

This formulation (1) underscores the impact of individual weights on the error metrics and guides the pruning process. While Lemma 3.1 simplifies the formulation for pruning a single weight, the general approach can be extended to multiple weights iteratively. This method facilitates a robust pruning strategy that is backed by both empirical results and theoretical foundations, bridging the gap in understanding observed in prior studies such as Wanda (Sun et al., 2023) and RIA (Zhang et al., 2024b).

**Corollary 3.2.** *Setting $\mathbf{Y} = \mathbf{0} \in \mathbb{R}^{c \times d}$ transitions our method to* input Wanda, *described by $\mathbf{S}_{jk} := |\mathbf{W}_{jk}|\|\mathbf{X}_{:j}\|_2$.*

This directly aligns with the objective in Sun et al. (2023), demonstrating that Wanda is a specific case under our broader framework.

**Corollary 3.3.** *Conversely, choosing $\mathbf{X} = \mathbf{0} \in \mathbb{R}^{a \times b}$ simplifies our pruning method to what we term* output Wanda *(denoted as OWanda), where the score matrix becomes $\mathbf{S}_{jk} := |\mathbf{W}_{jk}|\|\mathbf{Y}_{k:}\|_2$.*

**Corollary 3.4.** *By setting $\mathbf{X} = \mathbf{W}^\top \in \mathbb{R}^{c \times b}(a = c)$ and $\mathbf{Y} = \mathbf{W}^\top \in \mathbb{R}^{c \times b}(d = b)$, the score matrix $\mathbf{S}_{jk}$ is redefined as $|\mathbf{W}_{jk}|(\|\mathbf{W}_{j:}\|_2 + \|\mathbf{W}_{:k}\|_2)$.*

This configuration suggests an alternative masking approach and segues into a further analysis on how our method encompasses both Wanda and RIA as special cases. The following theorem provides a provable construction to recover the relative importance design in Zhang et al. (2024b).

**Theorem 3.5.** *Assuming $a = b$ and $c = d$, consider one of the following strategies:*

- $\mathbf{X}_{:j} := t_j(1; \dots; 1) \in \mathbb{R}^{b \times 1}$ *and* $\mathbf{Y}_{k:} := s_k(1, \dots, 1) \in \mathbb{R}^{1 \times c}$, *where* $t_j = (\sqrt{b}\|\mathbf{W}_{j:}\|_1)^{-1}$ *and* $s_k = (\sqrt{c}\|\mathbf{W}_{:k}\|_1)^{-1}$.

- $\mathbf{X} = \text{Diag}(\|\mathbf{W}_{1:}\|_1^{-1}, \dots, \|\mathbf{W}_{b:}\|_1^{-1})$ *and* $\mathbf{Y} = \text{Diag}(\|\mathbf{W}_{:1}\|_1^{-1}, \dots, \|\mathbf{W}_{:c}\|_1^{-1})$.

*For these configurations, the condition $\|\mathbf{X}_{:j}\|_2 + \|\mathbf{Y}_{k:}\|_2 = \alpha_{jk} := \|\mathbf{W}_{j:}\|_1^{-1} + \|\mathbf{W}_{:k}\|_1^{-1}$ holds for all $j, k$.*

This theorem elucidates that our methodology can invariably reconstruct the framework of relative importance RI in (Zhang et al., 2024b), validating the adaptability and breadth of our proposed pruning strategy.

## 3.3 From Relative Importance (RI) to RI Activation

In Theorem 3.5, we revisit the concept of Relative Importance (RI). Specifically, we represent RI by the following equation:

$$\mathbf{S}_{jk} = |\mathbf{W}_{jk}| \|\mathbf{W}_{j:}\|_1^{-1} + |\mathbf{W}_{jk}| \|\mathbf{W}_{:k}\|_1^{-1} \coloneqq \mathsf{RI}_{jk}.$$

Zhang et al. (2024b) also introduces an enhanced version of RI, termed RI with Activation (RIA), which incorporates the $\ell_2$-norm of activations:

$$\mathsf{RIA}_{jk} = \mathsf{RI}_{jk} \cdot \|\mathbf{X}_{:j}\|_2^{\alpha}, \tag{2}$$

where $\alpha$ is controlling the strength of activations.

This section aims to explore the derivation of RIA with theoretical grounding in RI. To clarify our notation and avoid confusion, we are aiming at finding the suitable $\mathbf{A} \in \mathbb{R}^{a \times b}$ and $\mathbf{B} \in \mathbb{R}^{c \times d}$ such as:

$$\|\mathbf{A}_{j:}\|_2 + \|\mathbf{B}_{:k}\|_2 = \left( \|\mathbf{W}_{j:}\|_1^{-1} + \|\mathbf{W}_{:k}\|_1^{-1} \right) \cdot \|\mathbf{C}_{:j}\|_2^{\alpha},$$

where $\mathbf{C}_{:j}$ will be instantiated as $\mathbf{X}_{:j}$ to satisfy Equation (2).

**Lemma 3.6.** *Let $p$ be a valid column index for $\mathbf{A}$. Define $\mathbf{A}_{uv} = 0$ for all $(u, v) \neq (j, p)$, and $\mathbf{A}_{j,p} = \|\mathbf{C}_{:j}\|_2^{\alpha} \|\mathbf{W}_{j:}\|_1^{-1}$. Similarly, let $s$ be a valid row index for $\mathbf{B}$. Define $\mathbf{B}_{uv} = 0$ for all $(u, v) \neq (s, k)$, and $\mathbf{B}_{s,k} = \|\mathbf{C}_{:j}\|_2^{\alpha} \|\mathbf{W}_{:k}\|_1^{-1}$. Then we recover Equation (2).*

The nonzero element in $\mathbf{A}$ ensures that the $\ell_2$-norm of the $j$-th row of $\mathbf{A}$ is: $\|\mathbf{A}_{j:}\|_2 = \|\mathbf{W}_{j:}\|_1^{-1} \cdot \|\mathbf{C}_{:j}\|_2^{\alpha}$. Similarly, the nonzero element in $\mathbf{B}$ ensures that the $\ell_2$-norm of the $k$-th column of $\mathbf{B}$ is: $\|\mathbf{B}_{:k}\|_2 = \|\mathbf{W}_{:k}\|_1^{-1} \cdot \|\mathbf{C}_{:j}\|_2^{\alpha}$. Combining these norms fulfills the intended equation.

## 3.4 General Solution

In Theorem 3.5, we presented two distinct strategies for recovering the relative importance as described in Zhang et al. (2024b). Following this, in Lemma 3.6, we constructed a method that accounts for both the weights and the input activations. Inspired by the diagonal design in Theorem 3.5, we now propose a general variant that considers both the weights and the activations.

Given that $\mathbf{D_X} \in \mathbb{R}^{b \times b}$ and $\mathbf{D_Y} \in \mathbb{R}^{c \times c}$ are diagonal matrices with entries defined as $(\mathbf{D_X})_{ii} = x_i = \|\mathbf{W}_{i:}\|_1^{-1}$ and $(\mathbf{D_Y})_{ii} = y_i = \|\mathbf{W}_{:i}\|_1^{-1}$ respectively, and $\mathbf{A} \in \mathbb{R}^{a \times b}$ and $\mathbf{B} \in \mathbb{R}^{c \times d}$ are arbitrary matrices, our objective is to compute the sum of norms: $\left\| (\mathbf{A} \mathbf{D_X})_{:j} \right\|_2 + \|(\mathbf{D_Y} \mathbf{B})_{k:}\|_2$.

**Lemma 3.7.** *Given the above definition, we show*

$$\left\| (\mathbf{A} \mathbf{D_X})_{:j} \right\|_2 + \|(\mathbf{D_Y} \mathbf{B})_{k:}\|_2 = \frac{\|\mathbf{A}_{:j}\|_2}{\|\mathbf{W}_{j:}\|_1} + \frac{\|\mathbf{B}_{k:}\|_2}{\|\mathbf{W}_{:k}\|_1}.$$

The utilization of the diagonal matrices $\mathbf{D_X}$ and $\mathbf{D_Y}$ simplifies the sum of the norms to the expressions derived above, offering insights into the influence of the weight matrix $\mathbf{W}$ on the norms of matrix transformations.

## 3.5 Enhanced Relative Importance Strategies

Beyond RIA, we propose several alternative strategies for relative importance that aim to minimize $\mathbf{S}_{jk}$ in Equation (1).

### 3.5.1 GENERALIZED $\ell_p$-NORM

Expanding beyond the conventional $\ell_1$-norm, we explore the utility of the $\ell_p$-norm in designing score matrices. In our approach, mirroring the strategy outlined in Theorem 3.5 for reconstructing RIA outcomes, we define the score as:

$$\mathbf{S}_{jk} = |\mathbf{W}_{jk}|(\|\mathbf{W}_{j:}\|_p^{-1} + \|\mathbf{W}_{:k}\|_p^{-1}). \tag{3}$$

Next, we are interested in finding the explicit formulation of $\mathbf{X}$ and $\mathbf{Y}$ instead of the norm representation when constructing the general $\ell_p$-norm.

**Lemma 3.8** (Generalized $\ell_p$-norm). *Let* $\mathbf{X}_{:j} = \|\mathbf{W}_{j:}\|_p^{-1} \cdot \|\mathbf{W}_{j:}\|_2^{-1} \cdot \mathbf{W}_{j:}^\top$ *and* $\mathbf{Y}_{k:} = \|\mathbf{W}_{:k}\|_p^{-1} \cdot \|\mathbf{W}_{:k}\|_2^{-1} \cdot \mathbf{W}_{:k}^\top$, *we recover Equation* (3).

Since the equation only requires $\|\mathbf{X}_{:j}\|_2 = \|\mathbf{W}_j\|_p^{-1}$, *any* vector with this $\ell_2$-norm will satisfy the condition. Inspired by this fact, we can consider the random unit vector scaling in the below lemma.

**Lemma 3.9** (Random unit vector scaling). *Choose any unit vector* $\mathbf{u}, \mathbf{v}$ *(i.e.,* $\|\mathbf{u}\|_2 = 1, \|\mathbf{v}\|_2 = 1$*) and set* $\mathbf{X}_{:j} = \|\mathbf{W}_{j:}\|_p^{-1} \cdot \mathbf{u}$ *and* $\mathbf{Y}_{k:} = \|\mathbf{W}_{:k}\|_p^{-1} \cdot \mathbf{v}$ *ensuring Equation* (3).

### 3.5.2 STOCHASTIC RELATIVE IMPORTANCE

Considering the computational and noise challenges associated with summing all elements across the full rows and columns of large matrices, we introduce a stochastic approach that involves sampling a subset of each row and column. This method assesses the effects of varying subset sizes, denoted by $\tau$, where $\tau < \min(b, c)$, on the overall performance. Specifically, we aim to:

a) Evaluate the sensitivity of the final performance to the size of $\tau$ when $\tau$ is reasonably large.

b) Determine if random sampling can enhance the results compared to a deterministic approach.

For this, we define the score matrix for a randomly sampled subset as:

$$\mathbf{S}_{jk} = |\mathbf{W}_{jk}|(\|\mathbf{W}_{j:S_j}\|_1^{-1} + \|\mathbf{W}_{S_k:k}\|_1^{-1}), \tag{4}$$

where $S_j$ and $S_k$ represent the sampled indices from the $j$-th row and $k$-th column, respectively, each with a cardinality of $\tau$. This approach builds on the RIA-inspired framework, adapting it for practical scenarios involving large-scale data.

For RIA in each weight layer, the reweighting sampling complexity is $O(b + c)$. In LLMs, $b$ and $c$ are always very large. Let's say the selection ratio is $\beta$, then for the stochastic relative importance design, the sampling complexity can be reduced to $O(\beta \min(b, c))$, which has been highly reduced.

**Lemma 3.10.** *Let* $S_j$ *and* $S_k$ *be index sets, and let* $\tau > 0$. *Define the vectors* $\mathbf{X}_{:j}$ *and* $\mathbf{Y}_{k:}$ *by*

$$\mathbf{X}_{:j}(i) = \frac{\mathbf{1}_{\{i \in S_j\}}}{\|\mathbf{W}_{j:S_j}\|_1 \sqrt{\tau}}, \quad \mathbf{Y}_{k:}(i) = \frac{\mathbf{1}_{\{i \in S_k\}}}{\|\mathbf{W}_{S_k:k}\|_1 \sqrt{\tau}}.$$

*Then these vectors satisfy Equation* (4).

### 3.6 TRAINING-FREE FINE-TUNING

We explore training-free fine-tuning within the context of the pruning-and-growing framework. Specifically, for the pruned weight matrix $\widetilde{\mathbf{W}}$, we aim to minimize the reconstruction error as defined in (Sym). Initially, we identify the growth index, followed by the pruning index, to maintain a consistent sparsity ratio. DSnoT (Zhang et al., 2023) developed a growing criterion based on the expected change in reconstruction error when reinstating a weight. Particularly, for any given weight row $q \in [1, b]$, the index $i$ is determined as follows:

$$i = \arg\max_r \text{sign}(\mathbb{E}[\epsilon_q]) \cdot \widetilde{\mathbf{W}}_{q,r} \cdot \mathbb{E}[\mathbf{X}_q]/\text{Var}(\mathbf{X}_q),$$

where $\epsilon_q \coloneqq \mathbf{W}_{q:}\mathbf{X} - \widetilde{\mathbf{W}}_{q:}\mathbf{X}$ denotes the reconstruction error of the $q$-th row across different input activations. It is important to note that for simplicity, output activations are not considered here,

which may provide an interesting avenue for future exploration. The functions $\text{sign}(\cdot)$, $\mathbb{E}[\cdot]$, and $\text{Var}(\cdot)$ denote the standard sign function, expectation, and variance of given inputs over $N \times L$ tokens, respectively. Drawing inspiration from the Wanda metric, the DSnoT model defines the pruning index $j$ as:

$$j = \underset{r:\Delta(q,r)<0}{\arg\min} |\widetilde{\mathbf{W}}_{q,r}| \|\mathbf{X}_q\|_2 ,$$

where $\Delta(q,r) := \text{sign}(\mathbb{E}[\epsilon_q]) \left(\widetilde{\mathbf{W}}_{q,r} \cdot \mathbb{E}[\mathbf{X}_q]\right)$.

Several simple yet effective modifications have been incorporated into the pruning-and-growing framework:

**a) Relative weight importance.** Both in determining the growing index $i$ and the pruning index $j$, we incorporate global information, emphasizing the relative importance of weights in neuron selection.

**b) Square root activation.** Our follow-up experiments on Wanda and RIA demonstrate the benefits of square root activation in determining the pruning index $j$.

**c) Regularized objective.** The method MagR (Zhang et al., 2024a) found that adding an $\ell_\infty$ norm helps reduce the magnitude of weights during quantization. Here, we adopt a more general regularizer, considering a general $\ell_p$ norm and focusing on specific rows rather than entire layers to reduce communication costs.

Define $\mathbf{D}_{q,r} := \|\widetilde{\mathbf{W}}_{q,:}\|_1^{-1} + \|\widetilde{\mathbf{W}}_{:,r}\|_1^{-1}$. The updated rule for identifying the growing index $i$ is formalized as:

$$i = \underset{r}{\arg\max} \left\{ \text{sign}(\mathbb{E}[\epsilon_q]) \cdot \mathbf{D}_{q,r} \cdot \frac{\mathbb{E}[\mathbf{X}_q]}{\text{Var}(\mathbf{X}_q)} + \gamma_1 \|\widetilde{\mathbf{W}}_q\|_p \right\}, \tag{5}$$

where $\gamma_1$ is the growing regularization parameter, striking a balance between fidelity and the $\ell_p$ regularizer. Similarly, the pruning index $j$ is now defined as:

$$j = \underset{r:\Delta(q,r)<0}{\arg\min} \left\{ |\widetilde{\mathbf{W}}_{q,r}| \cdot \mathbf{D}_{q,r} \cdot \|\mathbf{X}_q\|_2^\alpha + \gamma_2 \|\widetilde{\mathbf{W}}_q\|_p \right\}, \tag{6}$$

where $\Delta(q,r) := \text{sign}!(\mathbb{E}[\epsilon_q]), \left(\widetilde{\mathbf{W}}q,r \cdot \mathbf{D}q,r \cdot \mathbb{E}[\mathbf{X}_q]\right)$, and $\gamma_2$ denotes the pruning regularization parameter.

We name this approach *Relative and Regularized Dynamic Sparse No Training* ($R^2$-DSnoT). It enables efficient network fine-tuning without additional training, conserving computational resources while enhancing performance.

## 4 EXPERIMENTS

**Setup and configurations.** We assess the proposed methods across a broad spectrum of popular LLMs, including LlaMA2 (7b-13b) (Touvron et al., 2023b), LlaMA3-8b (Dubey et al., 2024), OPT-1.3b (Zhang et al., 2022a). We utilize publicly available model checkpoints from the HuggingFace Transformers library (Wolf et al., 2020) for our evaluations. Each experiment, focused on post-training pruning, is conducted on an NVIDIA A100-80G GPU. The effectiveness of each pruned model is primarily measured using the perplexity score on the Wikitext-2 dataset (Merity et al., 2016). For calibration, we use 128 samples from the C4 dataset (Raffel et al., 2020), with each sample comprising 2048 tokens. This approach ensures consistency with the settings used in baseline methods, enabling a fair comparison.

### 4.1 EFFICIENCY OF STOCHASTIC METHODS

We begin by examining two key designs discussed in Section 3.5: the generalized $\ell_p$ norm and stochastic relative importance. The results for the $\ell_p$ norm are presented in Appendix D.2, where we confirm that $p = 1$ is indeed optimal. We also compare various $\ell_p$ norm reweighting strategies,

Table 2: Perplexity comparison between StochRIA ($\beta=0.1$) and RIA on Wikitext-2 with $\alpha=1$. Mean $\pm$ std over 5 trials is shown for StochRIA; differences from RIA are in blue (better) and red (worse).

| Sparsity | Method | Sampling | LlaMA2-7b | LlaMA2-13b | LlaMA3-8b | OPT-1.3b |
|---|---|---|---|---|---|---|
| - | Dense | - | 5.47 | 4.88 | 6.14 | 14.62 |
| 50% | Magnitude | - | 16.03 | 6.83 | 205.44 | 1712.39 |
| | Wanda | - | 7.79 | 6.28 | 10.81 | 22.19 |
| | RIA | Full | 6.88 | 5.95 | 9.44 | 18.94 |
| | stochRIA | 10% | $6.91^{\pm0.0032}_{-0.03}$ | $5.95^{\pm0.0033}_{+0}$ | $9.46^{\pm0.025}_{-0.02}$ | $18.78^{\pm0.050}_{+0.16}$ |
| 2:4 | RIA | Full | 11.31 | 8.40 | 22.89 | 27.43 |
| | stochRIA | 10% | $11.41^{\pm0.046}_{-0.10}$ | $8.44^{\pm0.016}_{-0.04}$ | $23.74^{\pm0.230}_{+0.15}$ | $26.78^{\pm0.127}_{+0.65}$ |
| 4:8 | RIA | Full | 8.39 | 6.74 | 13.77 | 21.59 |
| | stochRIA | 10% | $8.44^{\pm0.014}_{-0.05}$ | $6.74^{\pm0.013}_{+0}$ | $13.93^{\pm0.095}_{-0.16}$ | $21.49^{\pm0.089}_{+0.10}$ |

Table 3: Perplexity scores on Wikitext-2 after training-free fine-tuning. The sparsity ratio is set to $60\%$ and $\alpha = 0.5$.

| Base | FT | LlaMA2-7b | LlaMA2-13b | LlaMA3-8b |
|---|---|---|---|---|
| Dense | - | 5.47 | 4.88 | 6.14 |
| Magnitude | - | 6.9e3 | 10.10 | 4.05e5 |
| Magnitude | DSnoT | 4.1e3 | 10.19 | 4.18e4 |
| Magnitude | $R^2$-DSnoT | **2.4e2** | **10.09** | **1.44e4** |
| Wanda | - | **9.72** | 7.75 | 21.36 |
| Wanda | DSnoT | 10.23 | **7.69** | 20.70 |
| Wanda | $R^2$-DSnoT | 10.08 | **7.69** | **20.50** |
| RIA | - | 10.29 | 7.85 | 21.09 |
| RIA | DSnoT | 9.97 | 7.82 | 19.51 |
| RIA | $R^2$-DSnoT | **9.96** | **7.78** | **18.99** |

with the results presented in Appendix D.3. Our primary focus, however, is on the findings related to stochastic relative importance, which, to the best of our knowledge, represents the first approach to incorporating stochasticity into LLM post-training pruning.

We analyze the impact of stochastic relative importance, with the results summarized in Table 2. The stochRIA results correspond to a sampling ratio of $\beta = 0.1$. Each reported value represents the mean performance across five trials with different random seeds. Notably, even with less than only 10% of the samples used to estimate relative importance, the results remain sufficiently representative, leading to promising outcomes.

In addition to unstructured pruning with a sparsity ratio of $0.5$, we also explore structured pruning using the N:M pattern (Zhou et al., 2021; Zhang et al., 2022b). The results are presented in Table 2. Noticed that here for intuitive comparison between RIA and stochRIA, we use the plain N:M structural pruning without channel permutation. These results consistently demonstrate the benefits and efficiency of our proposed method, stochRIA.

Furthermore, when aggregating results across all examined models and baselines, stochRIA achieves an accumulated perplexity that is $0.66$ lower than RIA, demonstrating the effectiveness of a stochastic design. This stochastic sampling preserves the diversity needed to handle subpopulations that rely on lower-average-importance weights while also helping preserve generalization by avoiding the dilution of salient features.

We also evaluate the performance across different sampling ratios, as shown in Appendix D.4. Our main takeaway is that stochRIA exhibits stable and competitive performance relative to RIA, particularly when the sampling ratio $\tau \geq 0.05$. At or above this threshold, the performance remains robust and occasionally surpasses less noisy sampling configurations. However, at an extremely low sampling ratio of $\tau = 0.01$, a significant performance drop is observed. Consequently, we adopt $\tau = 0.1$ as the default setting for our experiments.

## 4.2 TRAINING-FREE FINE-TUNING COMPARISONS

The intrinsic gap between pruned weights and the original, unpruned pretrained weights underscores the importance of minimizing reconstruction loss to achieve promising results. We introduced $R^2$-DSnoT, which incorporates relative weight reweighting and a regularized decision boundary during the dynamic sparse refinement step, all without additional training. Perplexity scores, as shown in Table 3, reveal that our $R^2$-DSnoT approach consistently surpasses baseline methods and the previous state-of-the-art DSnoT without fine-tuning. For instance, Magnitude exhibited subpar perplexity scores on LlaMA2-7b and LlaMA3-8b; however, our $R^2$-DSnoT achieved perplexity reductions of $96.5\%$ and $96.4\%$, respectively. These results not only validate $R^2$-DSnoT's efficacy but also offer guidance for scenarios involving high sparsity or underperforming pruned models, with minimal effort and no additional training.

**Zero-shot performance.** To provide a comprehensive evaluation, we also conducted zero-shot classification tests using seven well-regarded datasets. These tests assess the pruned models' ability to accurately categorize objects or data points into previously unseen categories. We employed

Table 4: Accuracies (%) for LLaMA2 models on 7 zero-shot tasks at 60% unstructured sparsity.

| Params | Method | BoolQ | RTE | HellaSwag | WinoGrande | ARC-e | ARC-c | OBQA | Mean |
|---|---|---|---|---|---|---|---|---|---|
| | Dense | 77.7 | 62.8 | 57.2 | 69.2 | 76.4 | 43.4 | 31.4 | 57.9 |
| LlaMA2-7b | Magnitude | 41.2 | 51.3 | 37.0 | 55.7 | 50.0 | 27.0 | 16.2 | 39.3 |
| | w. DSnoT | 43.2 | 54.2 | 38.4 | 56.4 | 53.3 | 27.7 | 20.6 | 41.1 |
| | w. $R^2$-DSnoT | 50.9 | 52.0 | 39.8 | 56.8 | 56.6 | 28.3 | 23.4 | **43.4** |
| | RIA | 66.1 | 53.1 | 43.5 | 63.2 | 64.6 | 30.2 | 26.0 | 49.5 |
| | w. DSnoT | 65.5 | 53.4 | 44.7 | 64.6 | 65.3 | 31.7 | 26.4 | 50.2 |
| | w. $R^2$-DSnoT | 65.2 | 53.8 | 44.7 | 65.1 | 65.0 | 31.6 | 27.0 | **50.3** |
| LlaMA3-8b | Dense | 81.3 | 69.7 | 60.1 | 73.0 | 80.1 | 50.4 | 34.8 | 64.2 |
| | Magnitude | 37.8 | 52.7 | 30.7 | 51.0 | 39.7 | 23.4 | 14.4 | 35.7 |
| | w. DSnoT | 37.8 | 52.7 | 33.4 | 49.9 | 43.5 | 23.0 | 14.8 | 36.4 |
| | w. $R^2$-DSnoT | 37.8 | 52.7 | 33.1 | 52.1 | 43.9 | 23.6 | 14.8 | **37.1** |
| | RIA | 70.2 | 53.4 | 39.7 | 61.7 | 61.1 | 28.6 | 20.4 | 47.9 |
| | w. DSnoT | 70.7 | 53.4 | 40.3 | 61.3 | 61.7 | 28.0 | 20.0 | 47.9 |
| | w. $R^2$-DSnoT | 70.4 | 53.4 | 40.3 | 61.9 | 61.2 | 28.3 | 21.0 | **48.1** |

the methodology described by Sun et al. (2023) and utilized tasks from the EleutherAI LM Harness (Gao et al., 2021), including BoolQ (Clark et al., 2019), RTE (Wang et al., 2018), HellaSwag (Zellers et al., 2019), WinoGrande (Sakaguchi et al., 2021), ARC (Easy and Challenge) (Clark et al., 2018), and OpenbookQA (Mihaylov et al., 2018). The results, presented in Table 4, show that $R^2$-DSnoT consistently outperforms DSnoT in zero-shot tasks, confirming its effectiveness. To the best of our knowledge, $R^2$-DSnoT establishes a new state-of-the-art for training-free pruning and fine-tuning methods in zero-shot performance.

## 5    DISCUSSION, LIMITATIONS, AND FUTURE WORK

This work introduced a unified symmetric formulation for LLM pruning, offering theoretical insight and strong empirical performance. Building on these findings, we outline several promising directions for future research:

**Beyond pruning.** Our exploration of Wanda and RIA introduced the symmetric objective in (Sym), initially aimed at post-training pruning for LLMs. However, our approach is extendable to post-training quantization and training-aware compression (Frantar et al., 2023; Egiazarian et al., 2024; Malinovskii et al., 2024), making these areas promising for future research.

**Better sampling.** In Section 4.1, we demonstrated that selective sampling of matrix rows and columns enhances both performance and efficiency by maintaining diversity in lower-importance weights. Future research could explore asymmetric or non-uniform sampling within the (Sym) framework to further optimize performance.

**Exploring symmetric designs.** As shown in Table 1, general and diagonal-specific symmetric designs for LLM compression highlight the potential of symmetric weight and activation patterns. Extending these approaches to distributed and federated settings (Yi et al., 2024; Ye et al., 2024) could also be valuable.

## 6    CONCLUSION

This study systematically analyzed post-training pruning methods, particularly Wanda and RIA, and provided both empirical evidence and theoretical insights into the role of input activations and relative weight importance, formalized through a unified symmetric objective in (Sym) that connects pruning with broader compression techniques. Building on this foundation, we proposed stochRIA, a stochastic variant that improves efficiency via selective sampling without compromising accuracy, and validated its effectiveness across various sparsity levels and model architectures. We further introduced a lightweight, training-free fine-tuning step within a prune-and-grow framework, achieving consistent improvements in perplexity and classification tasks over existing baselines. Together, these contributions advance both the theoretical understanding and practical utility of post-training pruning, and open up future directions in training-aware compression, quantization, and personalized deployment of large language models.

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

# CONTENTS

BROADER IMPACT

This work proposes a unified symmetric formulation and a set of practical algorithms for post-training pruning and compression of LLMs. By improving the efficiency of existing models without retraining, our methods lower the computational barrier for running LLMs, enabling broader accessibility in academic, industrial, and resource-constrained settings. This could benefit smaller research labs, educational institutions, and applications where deploying full-scale models is infeasible due to cost or hardware limitations.

On the positive side, this work contributes to democratizing access to powerful language models, potentially accelerating innovation in under-resourced regions and facilitating energy-efficient deployment on edge devices. In addition, our training-free fine-tuning approach promotes sustainability by reducing the need for compute-heavy finetuning procedures.

However, as with any work that enhances the deployability of LLMs, this research could also lower the barrier for misuse. Compressed models may be used in applications that propagate misinformation, generate spam, or amplify social biases embedded in the base models. Since our methods operate on publicly available LLMs, they inherit the original model's limitations and risks. We do not explicitly address fairness, robustness, or misuse detection in this work, and we encourage future research to consider safeguards, such as watermarking, usage monitoring, or alignment-aware pruning, to mitigate potential harms.

Overall, this work aims to improve the accessibility and computational efficiency of language models, while recognizing the importance of responsible deployment in real-world applications.

## A  MISSING PROOFS

### A.1  PROOF OF LEMMA 3.1

By using the definition of $g(\widetilde{\mathbf{W}})$ in Equation (InpRecon), we have

$$
g(\widetilde{\mathbf{W}}) = \sqrt{\sum_{k=1}^{c} \left\| \mathbf{X} \left( \widetilde{\mathbf{W}}_{:k} - \mathbf{W}_{:k} \right) \right\|_{2}^{2}} + \sqrt{\sum_{j=1}^{b} \left\| \left( \widetilde{\mathbf{W}}_{j:} - \mathbf{W}_{j:} \right) \mathbf{Y} \right\|_{2}^{2}}
$$

$$
= \sqrt{\sum_{k=1}^{c}\sum_{i=1}^{a} \left( \mathbf{X}_{i:} \left( \widetilde{\mathbf{W}}_{:k} - \mathbf{W}_{:k} \right) \right)^{2}} + \sqrt{\sum_{j=1}^{b}\sum_{l=1}^{d} \left( \left( \widetilde{\mathbf{W}}_{j:} - \mathbf{W}_{j:} \right) \mathbf{Y}_{:l} \right)^{2}}
$$

$$
= \sqrt{\sum_{k=1}^{c}\sum_{i=1}^{a} \left( \sum_{j=1}^{b} \mathbf{X}_{ij} \left( \widetilde{\mathbf{W}}_{jk} - \mathbf{W}_{jk} \right) \right)^{2}} + \sqrt{\sum_{j=1}^{b}\sum_{l=1}^{d} \left( \sum_{k=1}^{c} \left( \widetilde{\mathbf{W}}_{jk} - \mathbf{W}_{jk} \right) \mathbf{Y}_{kl} \right)^{2}}
$$

Now say we want to prune away just a single weight $\mathbf{W}_{jk}$. That is, we want to set $\widetilde{\mathbf{W}}_{jk} = 0$ and $\widetilde{\mathbf{W}}_{j'k'} = \mathbf{W}_{j'k'}$ for all $(j', k') \neq (j, k)$. For such a weight matrix $\widetilde{\mathbf{W}}_{jk}$ the expression for $f(\widetilde{\mathbf{W}})$ simplifies to

$$
g(\widetilde{\mathbf{W}}) = \sum_{i=1}^{a} \left( \sum_{j'=1}^{b} \mathbf{X}_{ij'} \left( \widetilde{\mathbf{W}}_{j'k} - \mathbf{W}_{j'k} \right) \right)^{2} + \sum_{l=1}^{d} \left( \sum_{k'=1}^{c} \left( \widetilde{\mathbf{W}}_{jk'} - \mathbf{W}_{jk'} \right) \mathbf{Y}_{k'l} \right)^{2}
$$

$$
= \sqrt{\sum_{i=1}^{a} \left( \mathbf{X}_{ij} \left( \widetilde{\mathbf{W}}_{jk} - \mathbf{W}_{jk} \right) + \sum_{j'\neq j} \mathbf{X}_{ij'} \left( \widetilde{\mathbf{W}}_{j'k} - \mathbf{W}_{j'k} \right) \right)^{2}}
$$

$$
+ \sqrt{\sum_{l=1}^{d} \left( \left( \widetilde{\mathbf{W}}_{jk} - \mathbf{W}_{jk} \right) \mathbf{Y}_{kl} + \sum_{k'\neq k} \left( \widetilde{\mathbf{W}}_{jk} - \mathbf{W}_{jk} \right) \mathbf{Y}_{kl} \right)^{2}}
$$

$$
= \sqrt{\sum_{i=1}^{a} (\mathbf{X}_{ij} (0 - \mathbf{W}_{jk}) + \sum_{j'\neq j} \mathbf{X}_{ij'} \underbrace{(\mathbf{W}_{j'k} - \mathbf{W}_{j'k})}_{=0})^{2}}
$$

$$
+ \sqrt{\sum_{l=1}^{d} ((0 - \mathbf{W}_{jk}) \mathbf{Y}_{kl} + \sum_{k'\neq k} \underbrace{\left( \widetilde{\mathbf{W}}_{jk} - \mathbf{W}_{jk} \right)}_{=0} \mathbf{Y}_{kl})^{2}}
$$

$$
= \sqrt{\sum_{i=1}^{a} (-\mathbf{X}_{ij} \mathbf{W}_{jk})^{2}} + \sqrt{\sum_{l=1}^{d} (-\mathbf{W}_{jk} \mathbf{Y}_{kl})^{2}}
$$

$$
= \sqrt{\sum_{i=1}^{a} \mathbf{X}_{ij}^{2} \mathbf{W}_{jk}^{2}} + \sqrt{\sum_{l=1}^{d} \mathbf{W}_{jk}^{2} \mathbf{Y}_{kl}^{2}}
$$

$$
= |\mathbf{W}_{jk}| \left( \|\mathbf{X}_{:j}\|_{2} + \|\mathbf{Y}_{k:}\|_{2} \right) \coloneqq \mathbf{S}_{jk}.
$$

### A.2  PROOF OF THEOREM 3.5

- Assume it is possible to choose matrices $\mathbf{X} \in \mathbb{R}^{a \times b}$ and $\mathbf{Y} \in \mathbb{R}^{c \times d}$ such that the identity

$$
\|\mathbf{X}_{:k}\|_{2} + \|\mathbf{Y}_{j:}\|_{2} = \alpha_{jk} \coloneqq \frac{1}{\|\mathbf{W}_{j:}\|_{1}} + \frac{1}{\|\mathbf{W}_{:k}\|_{1}} \tag{7}
$$

holds for all $j, k$. *This is always possible!*

Indeed, if we choose $a = b$, and let the $j$-th row of $\mathbf{X}$ be of the form $\mathbf{X}_{:j} := t_j(1; \cdots; 1) \in \mathbb{R}^{b \times 1}$, where $t_j = \frac{1}{\sqrt{b}\|\mathbf{W}_{j:}\|_1}$, then $\|\mathbf{X}_{j:}\|_2 = t_j \sqrt{b} = \frac{1}{\|\mathbf{W}_{j:}\|_1}$.

Similarly, if we choose $d = c$, and let the $k$-th column of $\mathbf{Y}$ be of the form $\mathbf{Y}_{:k} := s_k(1, \cdots, 1) \in \mathbb{R}^{1 \times c}$, where $s_k = \frac{1}{\sqrt{c}\|\mathbf{W}_{:k}\|_1}$, then $\|\mathbf{Y}_{:k}\|_2 = s_k \sqrt{c} = \frac{1}{\|\mathbf{W}_{:k}\|_1}$.

So, Equation (7) holds. In this case, our score matrix Equation (1) reduces to the plug-and-play method RIA (Zhang et al., 2024b).

- Another (even simpler) possiblity for constructing matrices $\mathbf{X}, \mathbf{Y}$ such that Equation (7) holds is as follows. Let $a = b$, and let $\mathbf{X} = \mathrm{Diag}(\|\mathbf{W}_{1:}\|_1^{-1}, \cdots, \|\mathbf{W}_{b:}\|_1^{-1})$. Clearly, for all $j = 1, \cdots, b$ we have $\|\mathbf{X}_{j:}\|_2 = \frac{1}{\|\mathbf{W}_{j:}\|_1}$.

  Similarly, let $d = c$, and let $\mathbf{Y} = \mathrm{Diag}(\|\mathbf{W}_{:1}\|_1^{-1}, \cdots, \|\mathbf{W}_{:c}\|_1^{-1})$. Clearly, for all $k = 1, \cdots, c$, we have $\|\mathbf{Y}_{:k}\|_2 = \frac{1}{\|\mathbf{W}_{:k}\|_1}$.

  Therefore, $\|\mathbf{X}_{:j}\|_2 + \|\mathbf{Y}_{k:}\|_2 = \frac{1}{\|\mathbf{W}_{j:}\|_1} + \frac{1}{\|\mathbf{W}_{:k}\|_1}$ for all $j, k$. So again, our score matrix (1) reduces to the plug-and-play method in Zhang et al. (2024b).

## A.3 PROOF OF LEMMA 3.7

Recall that in Section 3.4 $\mathbf{D_X} \in \mathbb{R}^{b \times b}$ and $\mathbf{D_Y} \in \mathbb{R}^{c \times c}$ are diagonal matrices with entries defined as $(\mathbf{D_X})_{ii} = x_i = \|\mathbf{W}_{i:}\|_1^{-1}$ and $(\mathbf{D_Y})_{ii} = y_i = \|\mathbf{W}_{:i}\|_1^{-1}$ respectively, and $\mathbf{A} \in \mathbb{R}^{a \times b}$ and $\mathbf{B} \in \mathbb{R}^{c \times d}$ are arbitrary matrices. We first compute $\mathbf{A} \mathbf{D_X}$. This product scales each column of $\mathbf{A}$ by the corresponding $x_i$. Specifically, for the $j$-th column, this operation is expressed as:

$$(\mathbf{A}\mathbf{D_X})_{:j} = x_j \mathbf{A}_{:j}.$$

The $\ell_2$-norm of this column is then given by:

$$\left\|(\mathbf{A}\mathbf{D_X})_{:j}\right\|_2 = x_j \|\mathbf{A}_{:j}\|_2 = \frac{\|\mathbf{A}_{:j}\|_2}{\|\mathbf{W}_{j:}\|_1}.$$

Next, we compute $\mathbf{D_Y} \mathbf{B}$. In this computation, each row of $\mathbf{B}$ is scaled by the corresponding $y_i$. For the $k$-th row, the scaling is represented as:

$$(\mathbf{D_Y}\mathbf{B})_{k:} = y_k \mathbf{B}_{k:}.$$

The $\ell_2$-norm of this row is:

$$\|(\mathbf{D_Y}\mathbf{B})_{k:}\|_2 = y_k \|\mathbf{B}_{k:}\|_2 = \frac{\|\mathbf{B}_{k:}\|_2}{\|\mathbf{W}_{:k}\|_1}.$$

Finally, we consider the sum of these norms:

$$\left\|(\mathbf{A}\mathbf{D_X})_{:j}\right\|_2 + \|(\mathbf{D_Y}\mathbf{B})_{k:}\|_2 = \frac{\|\mathbf{A}_{:j}\|_2}{\|\mathbf{W}_{j:}\|_1} + \frac{\|\mathbf{B}_{k:}\|_2}{\|\mathbf{W}_{:k}\|_1}.$$

The first term involves scaling the $j$-th column of $\mathbf{A}$ by $x_j$, with the resulting norm being the original column norm divided by the $\ell_1$-norm of the corresponding weights in $\mathbf{W}$. Similarly, the second term scales the $k$-th row of $\mathbf{B}$ by $y_k$, with the resulting norm also being the original row norm divided by the $\ell_1$-norm of the corresponding weights in $\mathbf{W}$.

## A.4 PROOF OF LEMMA 3.8

We aim to construct $\mathbf{X}_{:j}$ to be proportional to $\mathbf{W}_{j:}^\top$. A natural choice is to set

$$\mathbf{X}_{:j} = c \cdot \mathbf{W}_{j:}^\top,$$

where $c$ is a scalar to be determined. A similar condition applies when considering $\mathbf{Y}_{k:}$. The central task is to compute the corresponding scaling factor $c$ for both $\mathbf{X}$ and $\mathbf{Y}$.

To determine $c$, we choose it such that

$$\|\mathbf{X}_{:j}\|_2 = \left\|c \cdot \mathbf{W}_{j:}^\top\right\|_2 = \|\mathbf{W}_{j:}\|_p^{-1}.$$

We now compute the $\ell_2$-norm of $\mathbf{X}_{:j}$:

$$\left\|c \cdot \mathbf{W}_{j:}^\top\right\|_2 = |c| \cdot \left\|\mathbf{W}_{j:}^\top\right\|_2 = |c| \cdot \|\mathbf{W}_{j:}\|_2.$$

Setting this equal to $\|\mathbf{W}_{j:}\|_p^{-1}$, we have:

$$|c| \cdot \|\mathbf{W}_{j:}\|_2 = \|\mathbf{W}_{j:}\|_p^{-1}.$$

Solving for $c$, we obtain:

$$c = \frac{1}{\|\mathbf{W}_{j:}\|_p} \cdot \frac{1}{\|\mathbf{W}_{j:}\|_2}.$$

Using this value of $c$, we define $\mathbf{X}_{:j}$ as:

$$\mathbf{X}_{:j} = \frac{1}{\|\mathbf{W}_{j:}\|_p} \cdot \frac{1}{\|\mathbf{W}_{j:}\|_2} \cdot \mathbf{W}_{j:}^\top.$$

This construction ensures that

$$\|\mathbf{X}_{:j}\|_2 = \|\mathbf{W}_{j:}\|_p^{-1}.$$

Similarly, for $\mathbf{Y}$, we have:

$$\mathbf{Y}_{k:} = \frac{1}{\|\mathbf{W}_{:k}\|_p} \cdot \frac{1}{\|\mathbf{W}_{:k}\|_2} \cdot \mathbf{W}_{:k}^\top,$$

which satisfies Equation (3).

By combining these results, we conclude the proof of Lemma 3.8.

## A.5 PROOF OF LEMMA 3.9

Let $\mathbf{u}$ be any unit vector in $\ell_2$-norm, i.e., $\|\mathbf{u}\|_2 = 1$. Construct $\mathbf{X}_{:j} = \|\mathbf{W}_{j:}\|_p^{-1} \mathbf{u}$. Then by using the definition of the $\ell_2$-norm, we have

$$\|\mathbf{X}_{:j}\|_2 = \left\|\|\mathbf{W}_{j:}\|_p^{-1}\mathbf{u}\right\|_2 = \left|\|\mathbf{W}_{j:}\|_p^{-1}\right| \|\mathbf{u}\|_2 = \|\mathbf{W}_{j:}\|_p^{-1} \cdot 1 = \|\mathbf{W}_{j:}\|_p^{-1}.$$

Hence, we obtain $\|\mathbf{X}_{:j}\|_2 = \|\mathbf{W}_{j:}\|_p^{-1}$, which is exactly as desired.

Similarly, let $\mathbf{v}$ be any unit vector in $\ell_2$-norm, we have $|\mathbf{W}_{jk}| \cdot \|\mathbf{W}_{:k}\|_p^{-1}$.

Put them together, we prove Lemma 3.9.

## A.6 PROOF OF LEMMA 3.10

Given that $\mathbf{X}_{:j}$ and $\mathbf{Y}_{k:}$ are vectors to be constructed, $\mathbf{W}$ is a matrix, and $S_j$ and $S_k$ are randomly sampled index sets from the $j$-th row and $k$-th column of $\mathbf{W}$, respectively, each with cardinality $\tau$, our task is to construct $\mathbf{X}_{:j}$ and $\mathbf{Y}_{k:}$ with specific norms. Specifically, the goal is to construct $\mathbf{X}_{:j}$ and $\mathbf{Y}_{k:}$ such that:

$$\|\mathbf{X}_{:j}\|_2 + \|\mathbf{Y}_{k:}\|_2 = \frac{1}{\left\|\mathbf{W}_{j:S_j}\right\|_1} + \frac{1}{\left\|\mathbf{W}_{S_k:k}\right\|_1},$$

where $\mathbf{W}_{j:S_j}$ denotes the entries of the $j$-th row of $\mathbf{W}$ at indices in $S_j$, and $\mathbf{W}_{S_k:k}$ denotes the entries of the $k$-th column of $\mathbf{W}$ at indices in $S_k$.

We first define the support vector $\mathbf{e}_{S_j}$ of appropriate size (equal to the number of rows in $\mathbf{X}$) as:

$$(\mathbf{e}_{S_j})_i = \begin{cases} \frac{1}{\sqrt{\tau}}, & \text{if } i \in S_j, \\ 0, & \text{otherwise.} \end{cases}$$

The vector $\mathbf{e}_{S_j}$ has non-zero entries only at indices in $S_j$, each equal to $\frac{1}{\sqrt{\tau}}$, ensuring that the $\ell_2$-norm of $\mathbf{e}_{S_j}$ is 1:

$$\left\|\mathbf{e}_{S_j}\right\|_2 = \sqrt{\sum_{i \in S_j} \left(\frac{1}{\sqrt{\tau}}\right)^2} = \sqrt{\tau \cdot \left(\frac{1}{\sqrt{\tau}}\right)^2} = 1.$$

To construct $\mathbf{X}_{:j}$, we set:

$$\mathbf{X}_{:j} = \frac{1}{\left\|\mathbf{W}_{j:S_j}\right\|_1} \cdot \mathbf{e}_{S_j}.$$

A basic verification shows that the $\ell_2$-norm of $\mathbf{X}_{:j}$ is:

$$\left\|\mathbf{X}_{:j}\right\|_2 = \frac{1}{\left\|\mathbf{W}_{j:S_j}\right\|_1} \cdot \left\|\mathbf{e}_{S_j}\right\|_2 = \frac{1}{\left\|\mathbf{W}_{j:S_j}\right\|_1} \cdot 1 = \frac{1}{\left\|\mathbf{W}_{j:S_j}\right\|_1}.$$

Similarly, we define the support vector $\mathbf{e}_{S_k}$ of appropriate size (equal to the number of columns in $\mathbf{Y}$) as:

$$(\mathbf{e}_{S_k})_i = \begin{cases} \frac{1}{\sqrt{\tau}}, & \text{if } i \in S_k, \\ 0, & \text{otherwise.} \end{cases}$$

To construct $\mathbf{Y}_{k:}$, we set:

$$\mathbf{Y}_{k:} = \frac{1}{\left\|\mathbf{W}_{S_k:k}\right\|_1} \cdot \mathbf{e}_{S_k}^\top.$$

Adding the norms:

$$\left\|\mathbf{X}_{:j}\right\|_2 + \left\|\mathbf{Y}_{k:}\right\|_2 = \frac{1}{\left\|\mathbf{W}_{j:S_j}\right\|_1} + \frac{1}{\left\|\mathbf{W}_{S_k:k}\right\|_1},$$

which matches the desired expression.

**Alternative construction using $\ell_1$ and $\ell_2$ norms.**

By definition:

$$\left\|\mathbf{W}_{j:S_j}\right\|_1 = \sum_{i \in S_j} |w_{ji}|, \quad \left\|\mathbf{W}_{j:S_j}\right\|_2 = \sqrt{\sum_{i \in S_j} w_{ji}^2}.$$

We can construct $\mathbf{X}_{:j}$ as:

$$\mathbf{X}_{:j} = \frac{1}{\left\|\mathbf{W}_{j:S_j}\right\|_1} \cdot \frac{1}{\left\|\mathbf{W}_{j:S_j}\right\|_2} \cdot \mathbf{W}_{j:S_j}^\top,$$

where $\mathbf{W}_{j:S_j}^\top$ is a vector with entries:

$$(\mathbf{W}_{j:S_j}^\top)_i = \begin{cases} w_{ji}, & \text{if } i \in S_j, \\ 0, & \text{otherwise.} \end{cases}$$

Similarly, we can construct $\mathbf{Y}_{k:}$ as:

$$\mathbf{Y}_{k:} = \frac{1}{\left\|\mathbf{W}_{S_k:k}\right\|_1} \cdot \frac{1}{\left\|\mathbf{W}_{S_k:k}\right\|_2} \cdot \mathbf{W}_{S_k:k}^\top,$$

where $\mathbf{W}_{S_k:k}^\top$ is a vector with entries:

$$(\mathbf{W}_{S_k:k}^\top)_i = \begin{cases} w_{ik}, & \text{if } i \in S_k, \\ 0, & \text{otherwise.} \end{cases}$$

Putting everything together, we prove Lemma 3.10.

## B  SYMMETRIC WANDA VARIANT WITH SQUARED FROBENIUS NORMS

Choose $\varepsilon \in (0, 1]$. Given $\mathbf{X} \in \mathbb{R}^{a \times b}, \mathbf{W} \in \mathbb{R}^{b \times c}$ and $\mathbf{Y} \in \mathbb{R}^{c \times d}$, define

$$g'(\widetilde{\mathbf{W}}) := \|\mathbf{X}(\widetilde{\mathbf{W}} - \mathbf{W})\|_F^2 + \|(\widetilde{\mathbf{W}} - \mathbf{W})\mathbf{Y}\|_F^2,$$

and consider solving the problem

$$\text{mininimize } g'(\widetilde{\mathbf{W}}) \quad s.t. \quad \text{Mem}(\widetilde{\mathbf{W}}) \leq \varepsilon \text{Mem}(\mathbf{W}), \widetilde{\mathbf{W}} \in \mathbb{R}^{b \times c}.$$

Note that

$$g'(\widetilde{\mathbf{W}}) = \sum_{k=1}^{c} \left\| \mathbf{X}\left(\widetilde{\mathbf{W}}_{:k} - \mathbf{W}_{:k}\right) \right\|_2^2 + \sum_{j=1}^{b} \left\| \left(\widetilde{\mathbf{W}}_{j:} - \mathbf{W}_{j:}\right) \mathbf{Y} \right\|_2^2$$

$$= \sum_{k=1}^{c} \sum_{i=1}^{a} \left( \mathbf{X}_{i:} \left(\widetilde{\mathbf{W}}_{:k} - \mathbf{W}_{:k}\right) \right)^2 + \sum_{j=1}^{b} \sum_{l=1}^{d} \left( \left(\widetilde{\mathbf{W}}_{j:} - \mathbf{W}_{j:}\right) Y_{:l} \right)^2$$

$$= \sum_{k=1}^{c} \sum_{i=1}^{a} \left( \sum_{j=1}^{b} \mathbf{X}_{ij} \left(\widetilde{\mathbf{W}}_{jk} - \mathbf{W}_{jk}\right) \right)^2 + \sum_{j=1}^{b} \sum_{l=1}^{d} \left( \sum_{k=1}^{c} \left(\widetilde{\mathbf{W}}_{jk} - \mathbf{W}_{jk}\right) \mathbf{Y}_{kl} \right)^2$$

Now say we want to prune away just a single weight $\mathbf{W}_{jk}$. That is, we want to set $\widetilde{\mathbf{W}}_{jk} = 0$ and $\widetilde{\mathbf{W}}_{j'k'} = \mathbf{W}_{j'k'}$ for all $(j', k') \neq (j, k)$. For such a weight matrix $\widetilde{\mathbf{W}}_{jk}$ the expression for $g'(\widetilde{\mathbf{W}})$ simplifies to

$$g'(\widetilde{\mathbf{W}}) = \sum_{i=1}^{a} \left( \sum_{j'=1}^{b} \mathbf{X}_{ij'} \left(\widetilde{\mathbf{W}}_{j'k} - \mathbf{W}_{j'k}\right) \right)^2 + \sum_{l=1}^{d} \left( \sum_{k'=1}^{c} \left(\widetilde{\mathbf{W}}_{jk'} - \mathbf{W}_{jk'}\right) \mathbf{Y}_{k'l} \right)^2$$

$$= \sum_{i=1}^{a} \left( \mathbf{X}_{ij} \left(\widetilde{\mathbf{W}}_{jk} - \mathbf{W}_{jk}\right) + \sum_{j' \neq j} \mathbf{X}_{ij'} \left(\widetilde{\mathbf{W}}_{j'k} - \mathbf{W}_{j'k}\right) \right)^2$$

$$+ \sum_{l=1}^{d} \left( \left(\widetilde{\mathbf{W}}_{jk} - \mathbf{W}_{jk}\right) \mathbf{Y}_{kl} + \sum_{k' \neq k} \left(\widetilde{\mathbf{W}}_{jk} - \mathbf{W}_{jk}\right) \mathbf{Y}_{kl} \right)^2$$

$$= \sum_{i=1}^{a} (\mathbf{X}_{ij}(0 - \mathbf{W}_{jk}) + \sum_{j' \neq j} \mathbf{X}_{ij'} \underbrace{(\mathbf{W}_{j'k} - \mathbf{W}_{j'k})}_{=0})^2 + \sum_{l=1}^{d} ((0 - \mathbf{W}_{jk}) \mathbf{Y}_{kl} + \sum_{k' \neq k} \underbrace{\left(\widetilde{\mathbf{W}}_{jk} - \mathbf{W}_{jk}\right)}_{=0} \mathbf{Y}_{kl})^2$$

$$= \sum_{i=1}^{a} (-\mathbf{X}_{ij} \mathbf{W}_{jk})^2 + \sum_{l=1}^{d} (-\mathbf{W}_{jk} \mathbf{Y}_{kl})^2$$

$$= \sum_{i=1}^{a} \mathbf{X}_{ij}^2 \mathbf{W}_{jk}^2 + \sum_{l=1}^{d} \mathbf{W}_{jk}^2 \mathbf{Y}_{kl}^2$$

$$= \mathbf{W}_{jk}^2 \left( \|\mathbf{X}_{:j}\|_2^2 + \|Y_{k:}\|_2^2 \right) := \mathbf{S}_{jk}^2.$$

Our proposal is to choose entry $(j, k)$ which the smallest score $\mathbf{S}_{jk}$. Special cases:

1. If we choose $\mathbf{X} = \mathbf{0} \in \mathbb{R}^{a \times b}$, then our pruning method reduces to "output" Wanda:

$$\mathbf{S}_{jk} := |\mathbf{W}_{jk}| \, \|\mathbf{Y}_{k:}\|_2$$

2. If we choose $\mathbf{Y} = \mathbf{0} \in \mathbb{R}^{c \times d}$, then our pruning method reduces to "input" Wanda:

$$\mathbf{S}_{jk} := |\mathbf{W}_{jk}| \, \|\mathbf{X}_{:j}\|_2 \,.$$

3. If we choose $\mathbf{X} = \mathbf{W}^\top \in \mathbb{R}^{c \times b}(a = c)$ and $\mathbf{Y} = \mathbf{W}^\top \in \mathbb{R}^{c \times b}(d = b)$, then our score matrix becomes

$$\mathbf{S}_{jk} \overset{(27)}{=} |\mathbf{W}_{jk}| \sqrt{\|\mathbf{X}_{:j}\|_2^2 + \|\mathbf{Y}_{k:}\|_2^2} = |\mathbf{W}_{jk}| \sqrt{\|\mathbf{W}_{j:}\|_2^2 + \|\mathbf{W}_{:k}\|_2^2}$$

Letting $\mathbf{G}_{jk}^2 := \frac{1}{b+c} \left( \|\mathbf{W}_{j:}\|_2^2 + \|\mathbf{W}_{:k}\|_2^2 \right)$, note that

$$\begin{aligned}
\|\mathbf{G}\|_F^2 &= \sum_{j=1}^{b} \sum_{k=1}^{c} \mathbf{G}_{jk}^2 \\
&= \frac{1}{b+c} \sum_{j=1}^{b} \sum_{k=1}^{c} \left( \|\mathbf{W}_{j:}\|_2^2 + \|\mathbf{W}_{:k}\|_2^2 \right) \\
&= \frac{1}{b+c} \left( \sum_{j=1}^{b} \sum_{k=1}^{c} \|\mathbf{W}_{j:}\|_2^2 + \sum_{k=1}^{c} \sum_{j=1}^{b} \|\mathbf{W}_{:k}\|_2^2 \right) \\
&= \frac{1}{b+c} \left( c \sum_{j=1}^{b} \|\mathbf{W}_{j:}\|_2^2 + b \sum_{k=1}^{c} \|\mathbf{W}_{:k}\|_2^2 \right) \\
&= \frac{1}{b+c} \left( c\|\mathbf{W}\|_F^2 + b\|\mathbf{W}\|_F^2 \right) \\
&= \|\mathbf{W}\|_F^2
\end{aligned}$$

Clearly,

$$\frac{\mathbf{S}_{jk}^2}{(b+c)\|\mathbf{W}\|_F^2} = \frac{\mathbf{W}_{jk}^2 \mathbf{G}_{jk}^2}{\|\mathbf{W}\|_F^2}$$

4. Assume it is possible to choose matrices $\mathbf{X} \in \mathbb{R}^{a \times b}$ and $\mathbf{Y} \in \mathbb{R}^{c \times d}$ such that the identity

$$\sqrt{\|\mathbf{X}_{j:}\|_2^2 + \|\mathbf{Y}_{:k}\|_2^2} = \alpha_{jk} := \frac{1}{\|\mathbf{W}_{j:}\|_1} + \frac{1}{\|\mathbf{W}_{:k}\|_1}$$

holds for all $j, k$ (note that this is not always possible!). In this case, our score matrix reduces to the plug-and-play method of Zhang et al. (2024b).

## C INSIGHTS ON SENSITIVITY, ACTIVATION, AND SPARSITY

### C.1 COLUMN AND ROW SENSITIVITY

Compared with the Wanda design, RIA accounts for the relative importance of both rows and columns. However, it remains unclear whether columns and rows contribute equally to RIA's performance improvements. To investigate this, we conducted an extensive analysis of the significance of column-wise and row-wise relative importance, with the results shown in Table 5. A key finding is that the sum of the columns has more impact on performance, indicating greater importance.

Table 5: Perplexity scores on Wikitext-2, accounting for various norm $\alpha$ values and column & row sensitivity, with a sparsity ratio 50%.

| Model | LlaMA2-7b | | | | LlaMA2-13b | | | | LlaMA3-8b | | | | OPT-1.3b | | | |
|---|---|---|---|---|---|---|---|---|---|---|---|---|---|---|---|---|
| $\alpha$ | 0 | 0.5 | 1 | 2 | 0 | 0.5 | 1 | 2 | 0 | 0.5 | 1 | 2 | 0 | 0.5 | 1 | 2 |
| Dense | 5.47 | | | | 4.88 | | | | 6.14 | | | | 14.62 | | | |
| Wanda | 16.03 | 7.60 | 7.79 | 8.66 | 6.83 | 6.17 | 6.28 | 7.15 | 205.44 | 10.66 | 10.81 | 12.98 | 1712.39 | 22.14 | 22.19 | 24.74 |
| Col-Sum | 11.59 | 6.83 | 6.91 | 7.46 | 6.39 | **5.87** | 5.96 | 6.55 | 59.41 | 9.53 | 9.69 | 12.01 | 1062.66 | 18.28 | 18.41 | 22.25 |
| Row-Sum | 14.93 | 7.49 | 7.51 | 8.01 | 6.74 | 6.13 | 6.24 | 7.01 | 17.80 | 10.50 | 10.55 | 11.79 | 141.92 | 22.09 | 22.47 | 26.62 |
| RIA | 7.39 | **6.81** | 6.88 | 7.37 | 5.95 | 5.93 | 5.95 | 6.56 | 12.07 | **9.34** | 9.44 | 10.67 | 64.70 | **18.08** | 18.94 | 23.39 |

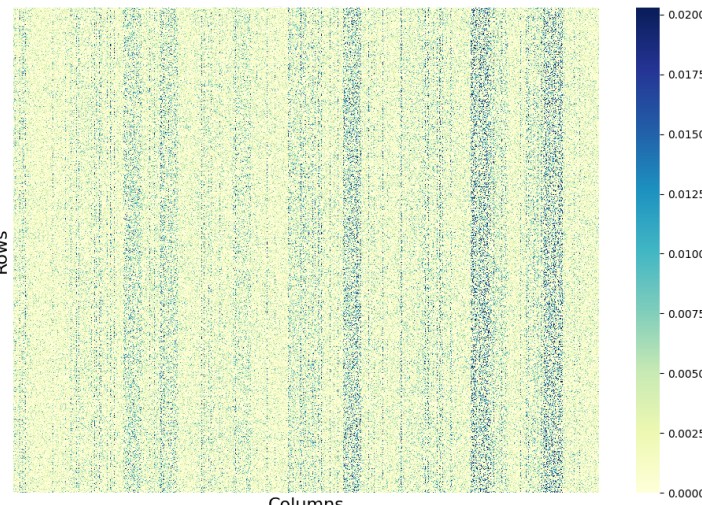

Figure 1: Visualization of the dense weight matrix in LLaMA2-7b.

To provide further insights, we visualized the heatmap of a randomly selected dense weight matrix from LLaMA2-7b, as illustrated in Figure 1. The heatmap displays stripe-like patterns, indicating column-specific structures where certain columns show significantly higher activations, forming distinct stripes. This observation suggests that normalizing by rows effectively balances these disparities. In cases where the rows within a specific column already exhibit relatively uniform distributions, normalization over rows may not be necessary. Thus, column normalization alone might suffice to balance the contributions of output neurons, especially when some columns dominate due to large absolute values.

## C.2 BENEFITS OF SQUARE ROOT INPUT ACTIVATION

In the design of Wanda (Sun et al., 2023), the power factor $\alpha$ applied to input activations is set to 1, whereas in RIA (Zhang et al., 2024b), $\alpha$ is adjusted to 0.5. In this study, we systematically explore the impact of varying the power factor on input activations, with detailed results presented in Table 5. An $\alpha$ value of 0 implies that no activation is considered in generating the pruning matrix. Our findings consistently show that incorporating input activation improves performance in terms of perplexity. Notably, $\alpha = 0.5$ proved optimal across various methods, underscoring the advantages of reducing the magnitude of input activations. We attribute this improvement to the mitigation of outliers in the input activations, where smoothing these values provides more meaningful guidance for pruning.

## C.3 VARIOUS UNSTRUCTURED SPARSITY RATIOS

We established a default unstructured sparsity ratio of 50%. In this section, we investigate the impact of varying sparsity ratios, as detailed in Table 6. For stochRIA, we report the mean average perplexity after three trials. Given that stochRIA has been shown to be stable, with variance examined in Table 1, we omit the variance to focus on performance. Our findings reveal that Wanda is

Table 6: Perplexity on Wikitext-2 with different sparsity. $\alpha = 1.0$.

| Sparsity | Method | Sampling | L2-7b | L2-13b | L3-8b | OPT-1.3b |
|---|---|---|---|---|---|---|
| Dense | - | - | 5.47 | 4.88 | 6.14 | 14.62 |
| 50% | Wanda | - | 7.79 | 6.28 | 10.81 | 22.19 |
| | RIA | Full | **6.88** | **5.95** | **9.44** | 18.94 |
| | stochRIA | 10% | 6.91 | **5.95** | 9.46 | **18.78** |
| 60% | Wanda | - | 15.30 | 9.63 | 27.55 | 38.81 |
| | RIA | Full | **10.39** | **7.84** | 19.52 | 26.22 |
| | stochRIA | 10% | 10.62 | 7.97 | **19.04** | **25.93** |
| 70% | Wanda | - | 214.93 | 104.97 | 412.90 | 231.15 |
| | RIA | Full | **68.75** | **51.96** | 169.51 | 98.52 |
| | stochRIA | 10% | 72.85 | 62.15 | **155.34** | **93.29** |

particularly sensitive to higher sparsity ratios, whereas both RIA and our proposed stochRIA demonstrate robustness to increased sparsity, maintaining stable performance across a broader range of conditions. Interestingly, we observed that on LLaMA3-8b and OPT1.3b, stochRIA consistently outperforms RIA, whereas on LLaMA2-7b and LLaMA2-13b, the reverse is true. This intriguing phenomenon may be attributed to the heavy noise present in the sampling process for LLaMA3-8b and OPT1.3b. In such cases, selecting a subset of weights through stochRIA may yield more reliable relative weight information, resulting in improved performance.

# D ADDITIONAL EXPERIMENTS

## D.1 IMPLEMENTATION DETAILS

Our selected baselines are implemented using the source code from Wanda[1] and RIA[2]. The default settings remain unchanged to ensure consistency. Notably, we explicitly set the sequence length to 2048 instead of using the maximum possible length to enable a fair comparison, following the strategy outlined in RIA.

The training-free fine-tuning component is based on DSnoT[3]. We configure the maximum cycle count to 50 and set the update threshold to 0.1. The default power of variance for regrowing and pruning is set to 1. Additionally, we incorporate the regularized relative design, resulting in our modified approach, DSnoT.

The seed for sampling the calibration data is set to 0. For N:M structural pruning, to enable an intuitive comparison, we use the standard approach without employing channel reallocation or linear sum assignment, as used in RIA.

## D.2 OPTIMAL $\ell_p$ NORM

In this study, we further explore the influence of the $\ell_p$ norm, considering standard norms where $p \in [1, 2, 3, 4]$, as well as the 0-norm and $\infty$-norm. The results are presented in Table 7. We observed that higher $p$ values degrade performance, as reflected by the perplexity scores, with $p = 1$ yielding the best results. This may be due to the fact that in pruning, significantly magnifying the differences between weights is not beneficial. Additionally, we found that both the 0-norm and $\infty$-norm do not yield promising results, as they capture only partial, and often highly biased, information about the weights.

---

[1] https://github.com/locuslab/wanda/tree/main
[2] https://github.com/biomedical-cybernetics/Relative-importance-and-activation-pruning
[3] https://github.com/zyxxmu/DSnoT

Table 7: Perplexity scores on Wikitext-2 for p-norm. The sparsity ratio is 50%, and all results correspond to $\alpha = 1$.

| p | LlaMA2-7b | LlaMA2-13b | LlaMA3-8b | OPT-1.3b |
|---|-----------|------------|-----------|----------|
| 1 | **6.88** | **5.95** | **9.44** | **18.95** |
| 2 | 6.90 | 5.96 | 9.48 | 19.02 |
| 3 | 6.95 | 6.01 | 9.57 | 19.66 |
| 4 | 7.12 | 6.08 | 9.92 | 20.77 |
| 0 | 7.78 | 6.28 | 10.81 | 22.17 |
| $\infty$ | 8.60 | 6.80 | 11.28 | 24.92 |

### D.3 $\ell_p$ NORM RE-WEIGHTING

In this section, we explore different $\ell_p$ norm re-weighting strategies. Our default re-weighting approach is defined in Equation (3) and is referred to as S1. Additionally, we investigate alternative strategies, denoted as S2, S3, and S4, as specified below:

$$S2 := \mathbf{S}_{jk} = |\mathbf{W}_{jk}|/(\|\mathbf{W}_{j:}\|_p + \|\mathbf{W}_{:k}\|_p),$$
$$S3 := \mathbf{S}_{jk} = |\mathbf{W}_{jk}| \cdot (\|\mathbf{W}_{j:}\|_p + \|\mathbf{W}_{:k}\|_p),$$
$$S4 := \mathbf{S}_{jk} = |\mathbf{W}_{jk}|/(\|\mathbf{W}_{j:}\|_p^{-1} + \|\mathbf{W}_{:k}\|_p^{-1}).$$

The comparative results for these strategies are presented in Table 8. As shown, our default strategy (S1) achieves the best performance, while the alternative designs fail to deliver improvements.

Table 8: Perplexity scores on Wikitext-2 for $\ell_p$-norm re-weighting with different strategies. The sparsity ratio is 50%, and all results are computed with $\alpha = 0.5$ and $p = 1$.

| Strategy | LLaMA2-7b | LLaMA2-13b | LLaMA3-8b | OPT-1.3b |
|----------|-----------|------------|-----------|----------|
| S1 (default) | 6.81 | 5.83 | 9.34 | 18.08 |
| S2 | 6.99 | 5.91 | 9.58 | 19.01 |
| S3 | 9.32 | 6.87 | 17.31 | 31.66 |
| S4 | 14.51 | 20.78 | 30.47 | 53.17 |

We hypothesize that the performance differences arise due to the relative magnitudes of the terms $\|\mathbf{W}_{j:}\|_p + \|\mathbf{W}_{:k}\|_p$ and $\|\mathbf{W}_{j:}\|_p^{-1} + \|\mathbf{W}_{:k}\|_p^{-1}$. Specifically, we assume that $\|\mathbf{W}_{j:}\|_p + \|\mathbf{W}_{:k}\|_p$ is typically large, while $\|\mathbf{W}_{j:}\|_p^{-1} + \|\mathbf{W}_{:k}\|_p^{-1}$ is generally small. Consequently, dividing by the former (S2) or multiplying by the latter (S4) reduces the magnitude of the pruning weights. We will provide statistical evidence to validate this assumption in subsequent sections.

### D.4 INFLUENCE OF SAMPLING RATIOS

In this section, we examine the impact of varying sampling ratios in stochRIA. It is important to note that these ratios are applied over $\min(b, c)$, where $b$ and $c$ represent the number of rows and columns in each layer, respectively. In Table 9, we can see the performance of stochRIA is generally stable and compares favorably to that of RIA when sampling across entire rows and columns, particularly for $\beta \geq 0.05$. At this threshold and above, the performance is robust, occasionally even surpassing less noisy sampling configurations. However, at an extremely low ratio of $\beta = 0.01$, there is a significant performance decline. Consequently, we have set $\beta = 0.1$ as the default setting for our experiments.

Table 10: $R^2$-DSnoT Hyperparameter Ablations on LLaMA3-8b. Each row shows the non-default hyperparameter values compared to the best-performing method.

| base | setting | $p$ | grow relative? | $\gamma_1$ | prune relative? | $\gamma_2$ | perplexity↓ |
|---|---|---|---|---|---|---|---|
| | best | 2 | ✓ | 0 | ✗ | 0.0001 | 18.99 |
| | $p$ | 1 | | | | | 19.04 |
| | | ∞ | | | | | 18.99 |
| Wanda | $\gamma$ | | | | | 0 | 18.99 |
| | | | | | | 0.001 | 18.99 |
| | relative | | ✗ | | ✗ | | 19.49 |
| | | | ✗ | | ✓ | | 19.25 |
| | | | ✓ | | ✓ | | 19.63 |
| | best | 2 | ✗ | 0 | ✓ | 0.001 | 20.50 |
| | $p$ | 1 | | | | | 25.61 |
| | | ∞ | | | | | 20.51 |
| RIA | $\gamma$ | | | | | 0 | 20.51 |
| | | | | | | 0.0001 | 20.52 |
| | relative | | ✗ | | ✗ | | 21.33 |
| | | | ✓ | | ✗ | | 22.16 |
| | | | ✓ | | ✓ | | 22.60 |

Table 9: Perplexity scores on Wikitext-2 for stochRIA with different sampling ratios. The sparsity ratio is 50%, and all results correspond to $\alpha = 1$. We highlight those performance drops over 0.1 as significant.

| ratio ($\beta$) | LlaMA2-7b | LlaMA2-13b | LlaMA3-8b | OPT-1.3b |
|---|---|---|---|---|
| 1 | 6.91 | 5.95 | 9.45 | 18.88 |
| 0.9 | 6.91 | 5.95 | 9.43 | 18.87 |
| 0.5 | 6.90 | 5.95 | 9.42 | 18.84 |
| 0.1 | 6.91 | 5.95 | 9.46 | 18.78 |
| 0.05 | 6.91 | 5.96 | 9.47 | 18.91 |
| 0.01 | 6.98 | 6.00 | 9.69 -0.24 | 19.36 -0.48 |

## D.5 ANALYSIS OF $R^2$-DSnoT HYPERPARAMETERS

In Section 3.6, we introduced the equations for our proposed $R^2$-DSnoT method, specifically Equation (5) and Equation (6). This method primarily involves three key hyperparameters: the regularization penalty $\gamma_1, \gamma_2$ and the norm type $p$. Additionally, we consider whether to apply relative importance reweighting during the growing or pruning phases—or during both. Given the number of hyperparameters, understanding their interactions can be computationally expensive and time-consuming.

To address this complexity, we adopt a systematic approach by performing a random search over 20 different combinations of hyperparameter settings. These combinations include: $p \in \{1, 2, \infty\}$, $\gamma_1 \in \{0, 0.0001, 0.001\}$, $\gamma_2 \in \{0, 0.0001, 0.001\}$, and binary choices for relative reweighting (True/False) during both the growing and pruning phases. For each of the 20 trials on the same model, we identify the best-performing combination and treat its hyperparameters as the "ground truth." We then evaluate the behavior under different scenarios and report the results in Table 10.

Our findings reveal several notable insights:

- Norm type $p$: The smooth $\ell_p$-norm with $p = 2$ consistently achieves the best performance. Compared to the non-differentiable $\ell_1$-norm, which underperforms due to its non-smooth nature, and the $\ell_\infty$-norm, which focuses only on the largest values and ignores smaller differences, the $\ell_p$-norm with $p = 2$ balances sensitivity and robustness effectively.

- Relative importance reweighting: Applying relative reweighting during either the growing or pruning phase improves performance significantly—yielding a 0.5 improvement on Wanda and 0.83 on RIA. However, applying reweighting to both phases simultaneously leads to substantial performance degradation, with a 0.64 and 2.1 drop on Wanda and RIA, respectively.

- Regularization penalty $\gamma$: The impact of $\gamma$ is minimal, as variations in its value result in only marginal differences in performance. This finding highlights the greater importance of the relative reweighting strategy.

