# OpenReview forum: "Symmetric Pruning for Large Language Models"
_ICLR.cc/2026/Conference — ICLR 2026 Conference Withdrawn Submission_

### Official Review · Reviewer_8M93 · 2025-10-29

**Soundness:** 2
**Presentation:** 1
**Contribution:** 2
**Rating:** 2
**Confidence:** 3

**Summary:**

This paper introduces an extended framework for post-training pruning, employing both calibration data in the input and output of a layer. They then use this framework to fit in several existing methods, such as Wanda and RIA, and determine the impact of pruning a single entry on the reconstruction error. They also introduce several new methods in this framework including OWanda, $l_p$ normalized RIA.

Next they define a subsampling based approach for the RIA method called StochRIA, and compare its performance on zero-shot classification, and perplexity w.r.t. Wanda and RIA.

Finally, they introduce several enhancements for the DSnoT methodology for training-free finetuning of the sparsity mask. These enhancements involve incorporating relative importance, regularizing, and taking the square root of the activation. This together results in a method that sets a new claimed state-of-the-art for training free pruning and fine-tuning.

**Strengths:**

**Originality**

The novel formulation of post-training pruning by incorporating output calibration allows for a significant contribution, showing the relationships between existing methods and can be used to develop new advanced post-training pruning methods.


**Significance**

The new claimed state-of-the-art with $R^2$ - DSnoT pushes the boundary on post-training pruning performance, which can lead to additional breakthroughs with the final goal of limiting the impact of post-training pruning.

Other pruning methods could derive from the proposed framework.

**Quality**

The experimental results are conducted on an extensive set of different tasks, models, and datasets showing the general applicability of the proposed solutions. Care is taken to use publicly available datasets, networks and baselines to ensure full reproducibility, and consistency with earlier papers.

The proofs of the introduced theorems (contained in the appendix) seem sound.

**Weaknesses:**

**Flow of the paper**

Table 1 (page 3) is significantly out of place. It serves as an overview of different post-training pruning methods in the proposed framework, but at that point in the paper, the general framework has not yet been introduced. This leads to a table with a lot of undefined (at that point) symbols, which combined with the numerous footnotes makes it incomprehensible without jumping back to it after reading the rest of the paper. As the first reference to this table is only made on line 470 (middle of page 9!), it would be much better to put this table at the end of the paper.

Line 375-376 state 'We begin by examining *two* key designs discussed in Section 3.5: *the generalized $l_p$ norm* and *stochastic relative importance*.' (emphasis mine) This would indicate that there are two main parts of the analysis, however directly afterwards the experiments on $l_p$ norm are relegated to different sections in the appendix, and only the stochastic relative importance is discussed in detail in the main paper.

The paper suggests several other masking approaches such as OWanda (using only the output calibration), and Symmetric or Generalized Diagonal approaches. These are however not expanded upon nor are they evaluated in the experiments section, serving as dead-ends.

**Notation**

Several symbols are either undefined, used multiple times in different contexts, or become defined later in the paper than their first occurrence. It would be best to define these symbols at their first occurrence, and minimize confusion between different values which use the same symbol.

* $C_{in}$, $C_{out}$ (line 148) are never defined. Likely these determine the number of input and output calibration samples? There is also a symbol $C_{:j}$ later on at line 236.
* $S_{jk}$ is first calculated on line 186, but is only named as 'Score Matrix' in line 201.
* $\alpha_{jk}$ is mentioned on line 214, but never occurs anywhere else. Additionally, this symbol conflicts with the $\alpha$ used to determine the power scaling of the activations in RIA.
* $S_j$, $S_k$ are used as the sampled indices from the j-th row, k-th column respectively. These symbols are virtually identical to the Score Matrix $S_{jk}$.

StochRIA is defined as a stochastic method to calculate Relative Importance, but importantly it does not incorporate activations in its pruning mechanism (as mentioned both in equation 4 on line 299, and footnote (e) of Table 1). However, its naming strongly suggests that it does as RIA stands for Relative Importance with **Activations**. Also different capitalizations (stochRIA and StochRIA) are used interchangeably throughout the paper.

**Applicability of the method**

Lines 405-406 mention 'Furthermore, when aggregating results across all examined models and baselines, stochRIA achieves an accumulated perplexity that is 0.66 lower than RIA, demonstrating the effectiveness of a stochastic design'
* For one it is not exactly clear whether you can simply accumulate the perplexity across different experiments, especially as they have variously different scales, and this fails to show the relative difference (which is a claimed 0.41% improvement over RIA when accumulating over all experiments).
* Secondly, this discrepancy is almost entirely based on the results of the OPT model which is the worst performing model (and then most of it is in the 2:4 sparsity regime), as for all other models the StochRIA approach has performance equivalent or worse to the RIA approach.
* Finally, this observation is dependent on an error in Table 2, where for the 2:4 sparsity setting on LlaMA3-8b the improvement of StochRIA is listed as +0.15, while in actuality it should be 22.89 - 23.74 = -0.85.

The specific time gain of StochRIA vs regular RIA is never explored.

$R^2$ - DSnoT is introduced as DSnoT with three additional improvements. However, in the main paper there is no ablation done on the different components that make up this method. Rather, this is relegated to the appendix, with no reference made to it in the main paper. As such, it is also not clear which combination of hyperparameters are exactly used in the experiments with $R^2$-DSnoT. At least, as indicated in section D.5, the regularization tradeoff has limited impact, so it can be questioned why this was included in the formulation.

**Coherence**

Throughout the experiments, two different sparsity ratios are used (50% in Table 2 and 60% in Tables 3,4). It would be best to either have a single unified sparsity, or to repeat each experiment with multiple sparsity ratios.

There seems to be a disconnect between two major parts of the paper, one which is related to the unified framework, and the other which is related to finetuning the mask.

**Minor textual mistakes**

Line 355 mentions a sign! function, this is never defined and likely a typo

The caption of Table 4 mentions only Llama2 models, but in the table itself also Llama3 models are used.

**Questions:**

Line 16-18 'This paper introduces new theoretical insights that redefine the standard minimization objective for pruning, offering a deeper understanding of the factors contributing to their success.'
* Which exactly are the new theoretical insights and the factors that contribute to the success of pruning?

Line 236. What exactly is the goal of defining a $C_{:j}$ if it will be instantiated as $X:j$ anyway?

On line 290-291 the paper mentions 'the computational and noise challenges associated with summing all elements across the full rows and columns of large matrices'
* What do you mean with the noise challenges?
* Can you quantify the actual computational challenges? What is the actual cost of RIA vs StochRIA in a real-world scenario? For inspiration, you could take a look at (Zhang2024) where in Section 5.4 they conduct a running time analysis of their approach w.r.t. existing approaches.

Line 337-338 'Our follow-up experiments on Wanda and RIA demonstrate the benefits of square root activation...'
* Which experiments does this setence refer to?
* I assume this is setting the $\alpha$ to 0.5 in equation 6?
* Has this not already been demonstrated for RIA?

Line 407-409 'This stochastic sampling preserves the diversity needed to handle subpopulations that rely on lower-average-importance weights while also helping preserve generalization by avoiding the dilution of salient features.'
* Can you actually prove or illustrate this claim?

Line 465-467 'We demonstrated that selective sampling of matrix rows and columns enhances both performance and efficiency by maintaining diversity in lower-importance weights'
* Similarly, can you actually prove or illustrate the claim that diversity is maintained?

What is the relationship between the selection ratio $\beta$ (lines 304-306) and the sampling ratio $\tau$ (lines 410-415)? They seem to be used interchangeably.

Line 1350-1353 'Applying relative reweighting during either the growing or pruning phase improves performance significantly .... However applying reweighting to both phases simultaneously leads to substantial performance degradation'
* Why is this the case that a combination of both fails? What causes relative growing to work well for one setting, while relative pruning works well for another setting?


**References**

(Zhang2024) : Zhang et al. 2024 'Plug-and-Play: An Efficient Post-training Pruning Method for Large Language Models'

---

### Official Review · Reviewer_2wmD · 2025-10-30

**Soundness:** 2
**Presentation:** 2
**Contribution:** 2
**Rating:** 4
**Confidence:** 4

**Summary:**

This paper introduces SymWanda, a unified theoretical framework for post-training pruning of LLMs. The framework minimizes reconstruction error by considering both inputs ($X$) and outputs ($Y$). The authors show that Wanda and RIA are special cases of this framework, and further derive new pruning strategies such as StochRIA, which improves efficiency via stochastic sampling. In addition, they propose $R^{2}$-DSnoT, a training-free fine-tuning method that helps recover pruned model performance. Experiments demonstrate that StochRIA and $R^{2}$-DSnoT achieve competitive results.

**Strengths:**

1. The paper introduces two novel and effective algorithms. StochRIA reduces computational cost via random sampling while maintaining or even improving performance. The training-free fine-tuning method $R^{2}$-DSnoT further enhances the performance of pruned models, demonstrating significant practical value.

2. The mathematical derivations for the proposed framework are logically sound.

**Weaknesses:**

1. The motivation for introducing the symmetric objective could be further clarified. In the original Wanda and RIA algorithms, $X$ represents true input activations, and its inclusion is well justified. However, the motivation for adding the output reconstruction term $||(\tilde{W}-W)Y||_F$ is unclear to me. A detailed discussion of why this term is necessary and how it improves upon the original input reconstruction (InpRecon) objective would strengthen the paper.

2. Furthermore, the practical implementation of $Y$ is not fully elaborated. On line 048, the paper states that SymWanda leverages both the input activation and output of each layer to minimize reconstruction error. However, in several variants listed in Table 1 (e.g., the Symmetric algorithm), both $X$ and $Y$ appear to be artificially constructed ($X=W^\top$, $Y=W^\top$) rather than derived from real data. A clearer explanation of what $Y$ represents in practice and why the construction (such as $W^\top$) are meaningful would be helpful.

3. While empirical results are shown for $l_p$-norm and StochRIA, there are no experiments for the other algorithms derived from the SymWanda framework (e.g., General Sym., OWanda, and Symmetric in Table 1). Including these results would significantly strengthen the evaluation, particularly in assessing whether incorporating $Y$ offers advantages over the original objective. For instance, compared with Wanda that performs pruning based solely on X, does the General Sym. method (incorporates both X and Y) demonstrate superior performance? I would be curious to hear the authors' views on this.

**Questions:**

1.	Could you please provide a more detailed explanation of the motivation for introducing the output reconstruction error term $||(\tilde{W}-W)Y||_F$?
2.	Could you explain what $Y$ represents in practice and why certain constructions (such as W^\top) are meaningful?
3.	Beyond showing that Wanda and RIA can be unified under the SymWanda framework, do Corollary 3.2 and Lemma 3.6 provide any deeper theoretical insights into these methods, as you mentioned on line 048-049?
4.	Could you provide experimental results for the other key variants derived from your framework, such as General Sym., OWanda, and Symmetric? This would be crucial for validating the practical utility of the SymWanda formulation itself.
5.	To support the efficiency claim, could you provide the actual runtime of the pruning process when using RIA and StochRIA (with $\beta = 0.1$)?

---

### Official Review · Reviewer_9B4w · 2025-10-31

**Soundness:** 3
**Presentation:** 4
**Contribution:** 2
**Rating:** 4
**Confidence:** 3

**Summary:**

This paper provides a mathematically unified view of existing pruning methods and introduces modest extensions. However, the contribution is theoretical rather than conceptual, and the empirical gains do not justify the complexity of the formulation.

**Strengths:**

1. Clear and systematic theoretical derivation: The authors unify several existing post-training pruning heuristics (Wanda, RIA, DSnoT) into a symmetric framework that includes both input and output activations, offering a cleaner mathematical view.
2. Training-free fine-tuning idea: The proposed R2-DSnoT provides a computationally cheap way to refine pruned models, which could be useful in resource-limited settings.

**Weaknesses:**

1. Incremental Contribution: The main “Symmetric” formulation largely re-derives existing pruning metrics in a generalized mathematical form. It is more a reinterpretation of Wanda/RIA than a new pruning principle. The resulting practical algorithms (StochRIA, R2-DSnoT) are also mild variations rather than fundamentally new ideas.
2. Unclear Practical Impact: Although the empirical results are consistent, the absolute improvements (≈0.5–1 perplexity or 1–2% zero-shot accuracy) are small. There is no analysis of runtime, FLOPs, or actual deployment gains, which limits the practical significance.
3. Limited Novelty in Fine-Tuning Component: R2-DSnoT extends DSnoT with a few heuristic terms (relative importance, lₚ regularizer), but lacks theoretical grounding or significant new behavior.
4. The theory section (Section 3) introduces many lemmas and corollaries, but these mainly restate known results in matrix norm terms. The connection between theory and actual pruning efficiency is weakly justified.

**Questions:**

1. How sensitive are the results to the choice of γ₁, γ₂, and the relative-importance scaling factor? Are these hyperparameters tuned per model, or shared across experiments?
2. Is R2-DSnoT remain stable under higher sparsity levels (e.g., 70–80%) or across different architectures (e.g., Mistral or Qwen)? The presented results seem limited to LLaMA-family models at 60% sparsity.

---

### Official Review · Reviewer_srD4 · 2025-11-01

**Soundness:** 2
**Presentation:** 3
**Contribution:** 3
**Rating:** 4
**Confidence:** 3

**Summary:**

This paper proposes a post-training pruning method for large language models based on a Symmetric Wanda formulation that balances input and output activations. It unifies prior pruning methods such as Wanda and RIA under a common symmetric objective and derives a closed-form single-weight removal cost to compute importance scores. Weights with the smallest scores are pruned to achieve the target sparsity. The method further introduces a normalization scheme to maintain consistency across layers. Finally, a training-free refinement step reweights and regularizes the pruned model, recovering performance without further training. Experiments on LLaMA and OPT models show that the method achieves better perplexity and zero-shot accuracy, especially when combined with the training-free refinement step.

**Strengths:**

- The paper provides a clear theoretical framework for understanding and unifying existing LLM post-training pruning methods, offering valuable analytical insight rather than relying purely on heuristics.

- It introduces a symmetric formulation that ensures scale invariance between input and output activations and defines pruning importance in a balanced and principled way through generalized $\ell$$_p$-norms.

- The work demonstrates strong analytical clarity, deriving closed-form expressions for weight importance and clearly articulating how each design choice affects pruning behavior and model fidelity.

**Weaknesses:**

- The symmetric pruning objective is theoretically well-defined for the $\ell$$_1$-based formulation, where the required condition can always be satisfied through proper construction of calibration matrices. However, because this linear formulation does not capture the nonlinear behavior of real LLMs (e.g., GeLU, softmax) or the true correlation between input and output activations, its practical representativeness is limited. Under the squared Frobenius norm, the same condition is not always possible to satisfy, further revealing a limitation of the symmetric assumption when extended to more realistic, nonlinear objectives.

- The paper does not examine the robustness of its pruning method under diverse calibration datasets. All experiments use 128 samples from the C4 dataset with fixed sequence length, and no evaluation is provided for domain shifts such as WikiText or OWT. This narrow calibration setup limits the assessment of generalization and stability across different data distributions.

- The proposed R2-DSnoT refinement introduces multiple hyperparameters, which add complexity and may require nontrivial tuning. Although the authors claim these parameters have little effect, their interaction can influence performance, and the method’s scalability to larger models or new datasets remains unclear.

**Questions:**

- How robust is the proposed symmetric pruning method to changes in calibration data distributions?

---

### Note · Authors · 2025-11-25

I have read and agree with the venue's withdrawal policy on behalf of myself and my co-authors.